# PerFit: Exploring Personalization Shifts in Representation Space of LLMs

**Jiahong Liu**[1],[*] **Wenhao Yu**[1], **Quanyu Dai**[2], **Zhongyang Li**[3], **Jieming Zhu**[2], **Menglin Yang**[4],
**Tat-Seng Chua**[5], **Irwin King**[1]
[1]CUHK   [2]Huawei   [3]Microsoft AI   [4]HKUST(GZ)   [5]NUS

## ABSTRACT

Personalization has become a pivotal field of study in contemporary intelligent systems. While large language models (LLMs) excel at general knowledge tasks, they often struggle with personalization, i.e., adapting their outputs to individual user expectations. Existing approaches that steer LLM behavior to meet users' implicit preferences and behavior patterns, primarily relying on tune-free methods (e.g., RAG, PAG) or parameter fine-tuning methods (e.g., LoRA), face challenges in effectively balancing effectiveness and efficiency. Moreover, the mechanisms underlying personalized preferences remain underexplored. To address these challenges, we first uncover key patterns of user-specific information embedded in the representation space. Specifically, we find that (1) personalized information lies within a low-rank subspace represented by vectors, and (2) these vectors demonstrate both a collective shift shared across users and a personalized shift unique to each individual user. Building on these insights, we introduce `PerFit`, a novel **two-stage solution that directly fine-tunes interventions in the hidden representation space** by addressing both collective and user-specific shifts, thereby achieving precise steering of LLM with minimal parameter overhead. Experimental results demonstrate that `PerFit` delivers strong performance across six datasets while **cutting the number of parameters by an average of 92.3%** compared to the state-of-the-art method. The code is available at the homepage.

## 1 INTRODUCTION

Large language models (LLMs) demonstrate remarkable abilities in text generation and complex reasoning (Radford et al.; Chang et al., 2024; Hu et al., 2024; Zhang et al., 2024d;c; Zhu et al., 2024; Wang et al., 2023; 2024a; Chen et al., 2026), thanks to comprehensive pre-training on diverse and large-scale datasets that equip them with broad general knowledge. Nonetheless, their optimization for wide-ranging tasks means they often struggle to adapt to individual user preferences. For instance, different users may expect distinct outputs even when given the same input. Accordingly, integrating user tastes and preferences into LLMs has propelled personalized large language models (PLLMs) to the forefront of research (Liu et al., 2025; Chen, 2023; Zhang et al., 2024e; Liu et al., 2024; Wang et al., 2024b). In real-world scenarios, user preferences are often implicit, like writing style and tone (Salemi et al., 2023; Tan et al., 2024b; Zhuang et al., 2024). Enabling LLMs to grasp this implicit information and generalize effectively to user queries remains a core research challenge for PLLMs.

Existing techniques can be broadly categorized into **tune-free methods**, such as retrieval-augmented generation (RAG) (Fan et al., 2024) and profile-augmented generation (PAG), and **parameter-efficient fine-tuning methods** (PEFT), like low-rank adaptation (LoRA) (Hu et al., 2021; Yang et al., 2024a). Non-tuned methods (Madaan et al., 2022; Salemi et al., 2023; Zhuang et al., 2024) emphasize efficiency and flexibility by leveraging external information or user profiles without modifying model parameters, but often struggle to achieve high personalization and generalization capability, especially when retrieved contexts contain noise that is misaligned with the user's real intent (Shi et al., 2023). In contrast, parameter fine-tuning methods (Tan et al., 2024b;a; Wagner et al., 2024; Qi et al., 2024)

---

[*]{jiahong.liu21}@gmail.com

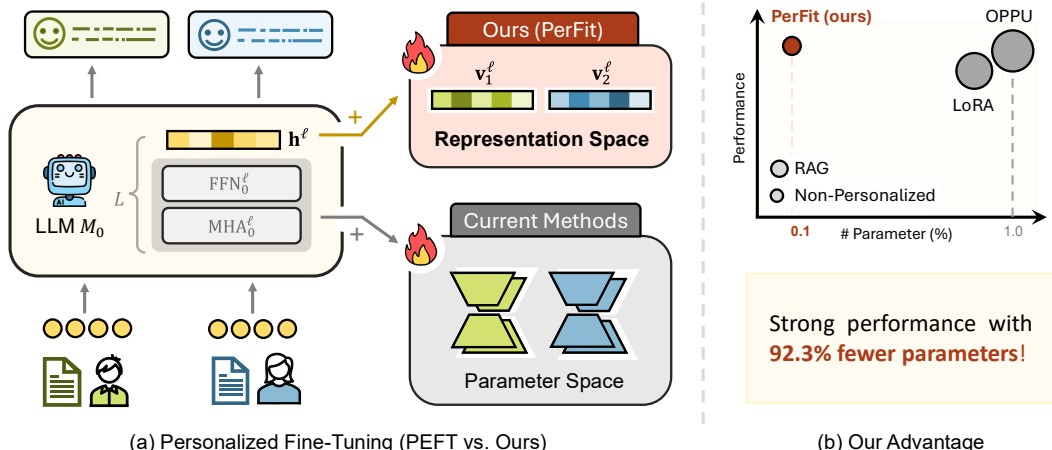

Figure 1: Illustration of our personalized fine-tuning method in representation space `PerFit`: (a) instead of tuning parameters, `PerFit` directly fine-tunes the hidden representations, where 🔥 represents fine-tuning with learnable parameters. (b) Experimental results show `PerFit` similarly strong performance on six datasets while reducing parameters by 92.3% on average compared to OPPU (Tan et al., 2024b).

update model parameters based on user data, enabling deeper and better personalization. Taking into account both model performance and the protection of user privacy, a prevalent approach is to allocate an individual PEFT module for each user (Tan et al., 2024b;a; Qi et al., 2024; Wagner et al., 2024; Gao & Zhang, 2024). LoRA still requires millions of parameters for good performance, though it reduces parameter counts (Wu et al., 2024; Cho et al., 2024; Guo et al., 2024). This leads to high communication costs and limited scalability in edge-cloud setups—where the base LLM runs in the cloud and personalized parameters reside on user devices such as phones (Qi et al., 2024; Wagner et al., 2024) (Appendix A). Therefore, **striking a balance between effectiveness and efficiency** remains a significant challenge for existing methods.

To solve this problem, we take the initial step of investigating how personalized information is captured by LLMs, thereby laying the groundwork for our alternative lightweight fine-tuning solution. This effort is motivated by recent advances in activation engineering (Wang et al., 2024d; Arditi et al., 2025; Turner et al., 2023; Zhang et al., 2024b), which allows precise control of LLM outputs by targeting internal representation interventions related to attributes like harmlessness (Bolukbasi et al., 2016; Park et al.), truthfulness (Li et al., 2023), and humor (Von Rütte et al., 2024). Therefore, the key question we investigate in this paper is:

> *Does personalized information induce discernible patterns in LLMs' hidden representation space that enable efficient guidance of model behavior?*

We conduct exploratory experiments to uncover personalized information encoded in the hidden representation space, named $\delta$-vectors (Section 3), revealing **two key observations**. (1) The $\delta$-vectors can be effectively represented within a low-dimensional orthogonal subspace (Observation 1). This suggests learning a **low-rank subspace** to get interventions representing user information in the representation space. (2) Vectors for all users in the low-rank subspace exhibit a clear **collective shift**, characterized by a common direction of deviation. Based on the collective shift, the vectors subsequently disperse towards multiple directions for different users (Observation 2). This suggests a two-stage approach to learn the collective and personalized shifts, respectively.

The intriguing findings inspire our personalized fine-tuning approach, which directly fine-tunes LLMs in the low-rank hidden representation subspace rather than model parameters, named `PerFit`. Specifically, we first train the collective shift using data from all users, and then, based on this, learn the personalized shifts for each user. To the best of our knowledge, **this is the first work to fine-tune LLMs in representation space tailored to personalized LLM tasks**. The learned collective shift, combined with the personalized shift, is directly added to the model's hidden representation space as an intervention to steer the model's output toward fulfilling individual users' personalized

requirements. Experimental results demonstrate that `PerFit` delivers strong performance across six personalization datasets while **cutting the number of parameters by an average of 92.3%** compared to the LoRA-based methods.

## 2 PRELIMINARY

### 2.1 PROBLEM STATEMENT

Let $\mathcal{U} = \{u_i\}_{i=1}^N$ be a set of $N$ users. Each user $u_i$ is associated with a set of input queries $\mathcal{Q}_i = \{q_j^{(i)}\}_{j=1}^{n_i}$ and corresponding desired outputs $\mathcal{Y}_i = \{y_j^{(i)}\}_{j=1}^{n_i}$, which implies the user's personalized preferences and expectations. Here, $n_i$ denotes the number of queries for user $u_i$. The base (i.e., non-personalized) LLM, denoted by $M_0$, generates generic outputs $\hat{y}_j^0 := M_0(q_j^{(i)})$ for any input query $q_j^i \in \mathcal{Q}_i$. Suppose $\Theta_0$ denotes the base model parameters and $\Theta_i$ denotes the parameters for user $u_i$, and the personalized parameters increment as $\Delta\Theta_i := \Theta_i \setminus \Theta_0$.

**Our objective** is to adapt $M_0$ into personalized models $M_i$ for each user $u_i$ such that for every $q_j^{(i)}$, the personalized output $\hat{y}_j^{(i)} = M_i(q_j^{(i)})$ closely matches the desired output $y_j^{(i)}$ **while minimizing parameter overhead** $|\Delta\Theta_i|$. Formally, this can be expressed as minimizing the aggregate loss:

$$\min_{\{M_i\}_{i=1}^N} \sum_{i=1}^N \sum_{j=1}^{M_i} \mathcal{L}\big(M_i(q_j^{(i)}), y_j^{(i)}\big),$$

where $\mathcal{L}(\cdot, \cdot)$ measures the discrepancy between model output and user target. This formulation encapsulates personalized fine-tuning of the base LLM to PLLM.

### 2.2 HIDDEN STATE REPRESENTATIONS AND ACTIVATION STEERING

**Hidden State Representation.** Our work concentrates on decoder-only transformer architectures (Liu et al., 2018). For the base model $M_0$, each layer $\ell \in L$ comprises the multi-head attention and feed-forward modules $\mathrm{MHA}_0^\ell$ and $\mathrm{FFN}_0^\ell$. Thus, the model can be expressed as: $M_0 = \bigcirc_{\ell \in L} \left(\mathrm{FFN}_0^\ell \circ \mathrm{MHA}_0^\ell\right)$, $\bigcirc$ denotes the composition of functions applied in sequence. The parameter set $\Theta_0$ is partitioned accordingly: $\Theta_0 = \bigcup_{\ell \in L} \left(\Theta_0^{\mathrm{MHA}^\ell} \cup \Theta_0^{\mathrm{FFN}^\ell}\right)$, where $\bigcup$ denotes the union of sets. The layer $\ell$ of the base model $M_0$ updates the hidden state $\mathbf{h}_t^\ell \in \mathbb{R}^d$ of the token $t$ as follows:

$$\mathbf{h}_t^{\ell+1} = \mathbf{h}_t^\ell + \mathrm{FFN}_0^\ell\big(\mathbf{h}_t^\ell + \mathrm{MHA}_0^\ell(\mathbf{h}_{1:t}^\ell)\big),$$

where $\mathrm{MHA}_0^\ell$ attends causally over tokens 1 through $t$, $d$ is the hidden dimension.

**Activation Steering.** Recent studies have explored how certain features are linearly represented in model hidden representation space utilizing activation steering (Tigges et al., 2023; Zhang et al., 2024b; Arditi et al., 2025), such as harmlessness (Bolukbasi et al., 2016; Park et al.), truthfulness (Li et al., 2023), and humor (Von Rütte et al., 2024). These feature directions serve as effective causal mechanisms, enabling precise control over model behavior and outputs via simple linear interventions. Activation steering adds an intervention (i.e., vector) $\mathbf{v}^\ell \in \mathbb{R}^d$ to the hidden state at layer $\ell$, modifying the model's behavior: $\tilde{\mathbf{h}}_t^\ell = \mathbf{h}_t^\ell + \mathbf{v}^\ell$. The next layer uses $\tilde{\mathbf{h}}_t^\ell$ instead of $\mathbf{h}_t^\ell$: $\mathbf{h}_t^{\ell+1} = \tilde{\mathbf{h}}_t^\ell + \mathrm{FFN}^\ell\big(\tilde{\mathbf{h}}_t^\ell + \mathrm{MHA}^\ell(\tilde{\mathbf{h}}_{1:t}^\ell)\big)$. This can be applied at any layer(s) to steer the model's output.

## 3 UNCOVERING PERSONALIZATION IN REPRESENTATION SPACE

Building on the insights of activation steering (Section 2.2), in this section, we aim to investigate *whether patterns related to personalized information exist within the hidden representation space.* If so, we can develop methods that leverage these patterns to guide personalization directly in representation space, *achieving a better balance between effectiveness and parameter efficiency.*

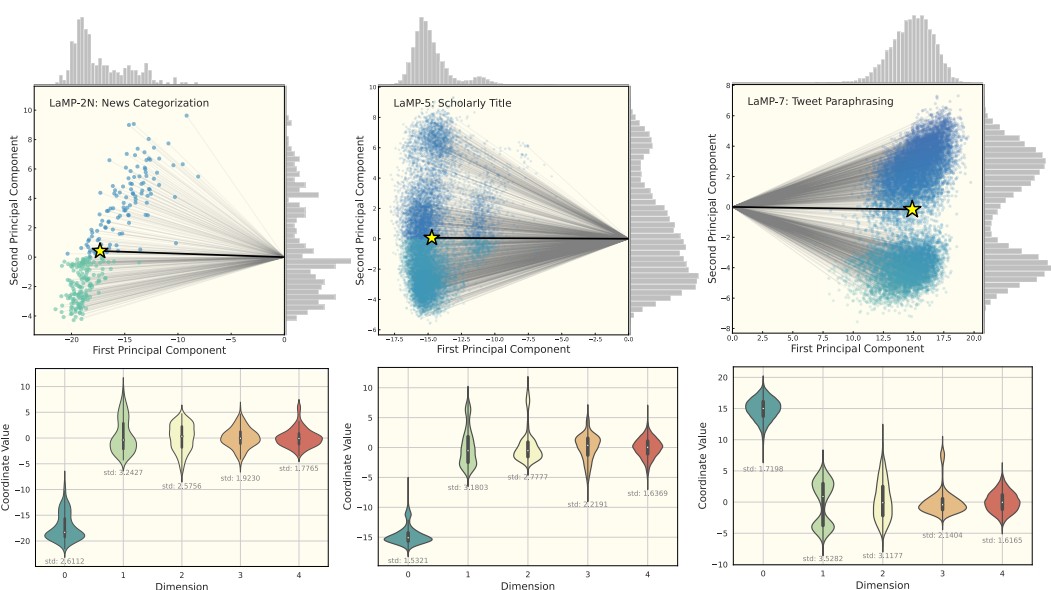

Figure 2: (1) The first row depicts the low-rank vector representations projected onto the first two principal components, with mean vectors indicated by yellow stars, demonstrating directional bias in the reduced-dimensional space. (2) The second row comprises violin plots of the coordinate value distributions across the first five feature dimensions. Here, the zeroth dimension shows a significant mean shift from zero, indicating a shared directional bias (i.e., collective shift) among users, whereas the remaining dimensions have means near zero with relatively large standard deviations, reflecting individual user variability (i.e., personalized shifts).

## 3.1 Extracting personalization vectors

Following the analysis paradigm of activation engineering (Arditi et al., 2025), for each user $u_i \in \mathcal{U}$, given their original query set $\mathcal{Q}_i^{\text{orig}}$, we enhance each query by incorporating the most relevant personalized information. The resulting personalization-enhanced query set is denoted as $\mathcal{Q}_i^{\text{per}}$. At layer $\ell$, let $\mathbf{h}_t^{(\ell)}(q) \in \mathbb{R}^d$ be the hidden state (residual stream activation) corresponding to the last token $t$ of the input query $q$. The mean residual representations for the original and personalized inputs are defined as $\boldsymbol{m}_i^{(\ell)} = \frac{1}{|\mathcal{Q}_i^{\text{orig}}|} \sum_{q \in \mathcal{Q}_i^{\text{orig}}} \mathbf{h}_t^{(\ell)}(q), \boldsymbol{n}_i^{(\ell)} = \frac{1}{|\mathcal{Q}_i^{\text{pers}}|} \sum_{q \in \mathcal{Q}_i^{\text{pers}}} \mathbf{h}_t^{(\ell)}(q)$.

The *difference-in-means* (Belrose, 2024) personalization vector at the layer $\ell$ is then $\mathbf{v}_i^{(\ell)} = \boldsymbol{n}_i^{(\ell)} - \boldsymbol{m}_i^{(\ell)}$, which captures the principal change in the model's internal representation induced by personalized information of user $i$. **Note that personalized information, unlike clear-cut traits such as harmlessness or helpfulness that can be manipulated via a single vector, is inherently more complex and diverse.** Therefore, we consider each user a special personality and analyze all users together to capture both collective and personalized aspects. The collection of $\mathbf{v}_i^{(\ell)}$ for all users $i \in \mathcal{U}$ is called $\delta$-**vectors** in this paper for simplicity. More details are shown in Appendix B.1.

Table 1: Minimum feature dimensions $r$ needed to explain 0.8 and 0.9 of variance. ‰ represents the $r$ as a per mille of total dimensions.

|         | 0.8 | | 0.9 | | 0.95 | |
|---------|-----|-----|-----|------|------|-------|
|         | $r$ | ‰   | $r$ | ‰    | $r$  | ‰     |
| LaMP-2N | 1   | 3.65 | 4  | 14.60 | 20  | 72.90 |
| LaMP-5  | 4   | 0.98 | 40 | 9.77  | 203 | 49.56 |
| LaMP-7  | 3   | 0.73 | 32 | 7.81  | 177 | 43.21 |

## 3.2 Observations

Based on the $\delta$-vectors, which isolate the personalized information of all users, we proceed to uncover the underlying personalization patterns. Below are the key observations.

> **Observation 1** (Low-rank Subspace). *The $\delta$-vectors can be effectively represented within a low-dimensional orthogonal subspace, significantly reducing the original feature space dimensionality.*

We performed singular value decomposition (SVD) (Stewart, 1993) on the obtained $\delta$-vectors to determine the intrinsic rank required to represent them with minimal loss of information. Table 1 reveals that the effective rank is significantly lower than the full dimensionality of the feature matrix, accounting for approximately 0.073% of the original dimensions. This observation suggests that the $\delta$-vectors vectors lie predominantly within a low-dimensional orthogonal subspace, suggesting substantial redundancy in the high-dimensional representations.

> **Observation 2** (Collective and Personalized Shifts). *The $\delta$-vectors exhibit a collective shift, accompanied by personalized shifts reflecting individual variability.*

We further plotted the mean and standard deviation of each dimension within the low-rank subspace based on the SVD. As shown in Figure 2, there is a significant shift with small variance in the low-rank subspace, indicating a collective shift across all vectors.

## 4 METHODOLOGY: PerFit

These findings have practical implications: understanding and isolating personalized representations enables the development of more efficient, lightweight fine-tuning methods with reduced computational demand. Leveraging the observations, the personalized method PerFit is proposed to **directly fine-tune the representation low-rank subspace and the intervention vector**, rather than the model parameters. Inspired by the representation fine-tuning paradigm (Wu et al., 2024), we propose a novel two-stage formulation specifically designed to achieve the personalization goal [1].

> **PerFit — Personalized Fine-Tuning in Representation Space [Algorithm 1]**
>
> $$\Phi_{\text{PerFit}}(\mathbf{h}) = (\phi_{\Delta\Theta^{(2)}} \circ \phi_{\Delta\Theta^{(1)}})(\mathbf{h}), \tag{1}$$
>
> where for $s = 1, 2$,
>
> $$\phi_{\Delta\Theta^{(s)}} : \mathbb{R}^d \to \mathbb{R}^d, \quad \phi_{\Delta\Theta^{(s)}}(\mathbf{x}) := \mathbf{x} + \underbrace{\mathbf{R}^{(s)\top}\left(\mathbf{W}^{(s)}\mathbf{x} + \mathbf{b}^{(s)} - \mathbf{R}^{(s)}\mathbf{x}\right)}_{\text{intervention vector } \mathbf{v}^{(s)}}. \tag{2}$$
>
> Here, $\mathbf{h}, \mathbf{x} \in \mathbb{R}^d$ and $\circ$ is the functional composition, and $\Delta\Theta^{(s)} = (\mathbf{R}^{(s)}, \mathbf{W}^{(s)}, \mathbf{b}^{(s)})$ are trainable parameter sets with $\mathbf{R}^{(s)}, \mathbf{W}^{(s)} \in \mathbb{R}^{r_s \times d}$, $\mathbf{b}^{(s)} \in \mathbb{R}^{r_s}$, where, $r_s \ll d$, $\mathbf{R}^{(s)}$ is a row-wise orthogonal matrix satisfying $\mathbf{R}^{(s)}(\mathbf{R}^{(s)})^\top = \mathbf{I}_{r_s}$.

**Intuitive Explanation.** PerFit is designed to align with our key observations (Figure 3).

- $\mathbf{R}$ is an orthogonal matrix that projects vectors from a high-dimensional space onto a low-dimensional subspace, consistent with the **low-rank subspace observation** (Observation 1). Its transpose, $\mathbf{R}^\top$, performs the inverse mapping by projecting vectors from the low-dimensional subspace back to the original high-dimensional space. The intervention vector $\mathbf{v}$ corresponds to the $\delta$-vectors , and the model directly learns these vectors during training.

- Two-stage fine-tuning functions $\phi_{\Delta\Theta^{(2)}} \circ \phi_{\Delta\Theta^{(1)}}$ are designed based on Observation 2 that $\Delta\Theta^{(1)}$ is tuned by all users' data $\mathcal{U}$ to get the **collective shift** for the first stage. Then, we fine-tune $\Delta\Theta_i^{(2)}$ for each user $u_i \in \mathcal{U}$ to get the **personalized shifts**. More explanations and analysis are shown in Appendix C .

---

[1] For simplicity, we remove the layer index $\ell$ in the notation.

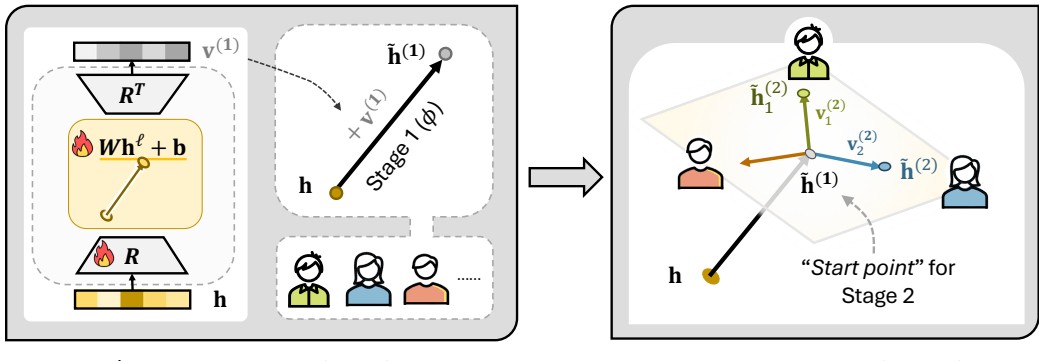

(a) **Stage 1**: Collective shift       (b) **Stage 2**: Personalized shifts

Figure 3: Illustration of two-stage personalized fine-tuning `PerFit`. (a) The first stage tunes on all users to obtain the collective shift. (b) The second-stage intervention vector is learned from the intervened representation of stage 1 and fine-tuned individually for each user.

In addition to the intuitive design guided by our observations, we theoretically show that LoRA-based methods, due to structural constraints of the framework, **cannot in general achieve the desired formulation** of collective and personalized representation shifts. (Details are in Appendix C.3).

## 5 EXPERIMENTS

We conduct extensive experiments to evaluate our proposed `PerFit` method across six diverse tasks from the LaMP benchmark. Our evaluation mainly focuses on the following three research questions:

- **RQ1.** How does `PerFit` perform compared to state-of-the-art personalized approaches in terms of both effectiveness and efficiency? (Section 5.2)

- **RQ2.** To what extent does `PerFit` improve computational and memory efficiency while maintaining competitive performance? (Section 5.3)

- **RQ3.** How does our two-stage training approach contribute to the model's performance, and what is the impact of each stage? (Section 5.4)

Beyond the primary research questions, we further examine several supplementary aspects of our method, such as stability, extended ablation studies, and performance in cold-start scenarios. Comprehensive details and results of these analyses are presented in Appendix E.

### 5.1 EXPERIMENTAL SETUP

This section outlines the experimental settings for evaluating our proposed `PerFit` method. We describe the datasets, baseline models, and key implementation parameters used in our evaluation. For additional setup details, please refer to the corresponding subsections in Appendix D.

**Datasets.** We conduct experiments on six diverse tasks from the LaMP benchmark (Salemi et al., 2024b): three classification tasks (News Categorization, Movie Tagging, Product Rating) and three generation tasks (News Headline Generation, Scholarly Title Generation, Tweet Paraphrasing). Following established practices (Tan et al., 2024b), data from approximately 100 users with the most extensive interaction histories for each task constitute our test set, while the remaining data is used for training the base (i.e., non-personalized) LLM. Details are provided in Appendix D.1.

**Baselines.** `PerFit` is compared against a range of baselines, all implemented using Llama2-7B as the base model. These include *Non-Tuned Methods*: **Non-Personalized**, Profile Augmented Generation (**PAG**) (Richardson et al., 2023), Retrieval Augmented Generation (**RAG**) (Salemi et al., 2024b) (with $k \in \{1, 2, 4\}$ retrieved documents), and **StyleVector** (Zhang et al., 2025); and *Tuned Methods*: Collective LoRA (Hu et al., 2021) (**LoRA-C**), Personalized LoRA (**LoRA-P**), **LoFiT** (Yin et al., 2024), **OPPU** (Tan et al., 2024b). Details of each baseline are available in Appendix D.3. The implementation details are in Appendix D.4 and full specifics on hyperparameters in Appendix D.5.

## 5.2 MAIN RESULTS (RQ1)

We evaluate `PerFit` against state-of-the-art personalized approaches across six diverse tasks from the LaMP benchmark. The results are presented in Tables 2 and 3, which demonstrate the effectiveness of our method in both personalized classification and generation scenarios[2].

Table 2: Results on classification tasks. We report Accuracy (Acc) and F1 Score (F1) for LaMP-2N and LaMP-2M, Mean Absolute Error (MAE) and Root Mean Squared Error (RMSE) for LaMP-3. For `PerFit`, we show the parameter percentage relative to the total model and the parameter reduction compared to OPPU. Blue and red numbers represent Stage-1 and Stage-2 parameters, respectively.

| Method | News Categorization (LaMP-2N) | | Movie Tagging (LaMP-2M) | | Product Rating (LaMP-3) | |
|---|---|---|---|---|---|---|
| | Acc ↑ | F1 ↑ | Acc ↑ | F1 ↑ | MAE ↓ | RMSE ↓ |
| LoRA-C | 0.787 | 0.538 | 0.478 | 0.425 | 0.223 | 0.491 |
| LoRA-P | 0.591 | 0.397 | 0.528 | 0.383 | 0.183 | 0.502 |
| LoFiT | 0.758 | 0.525 | 0.566 | 0.440 | TLE | TLE |
| OPPU | 0.810 | **0.589** | 0.600 | 0.493 | **0.179** | **0.443** |
| Ours (**PerFit**) | **0.818** | 0.586 | **0.630** | **0.518** | **0.179** | **0.443** |
| *- Param. Percentage ↓ (%)* | 0.0058 | 0.0117 | 0.0078 | 0.0010 | 0.0117 | 0.0015 |
| *- Param. Reduction ↑ (%)* | 93.75 | 81.25 | 91.67 | 98.44 | 87.50 | 97.66 |

Table 3: Results on generation tasks. We report ROUGE-1 (R-1) and ROUGE-L (R-L) metrics for LaMP-4, LaMP-5, and LaMP-7 tasks. The table compares both *Non-Tuned Methods* and *Tuned Methods* to demonstrate the effectiveness of different personalization approaches. For `PerFit`, we show the parameter percentage relative to the total model size and the parameter reduction compared to OPPU. Blue and red numbers represent Stage-1 and Stage-2 parameters respectively.

| Method | News Headline Gen. (LaMP-4) | | Scholarly Title Gen. (LaMP-5) | | Tweet Paraphrasing (LaMP-7) | |
|---|---|---|---|---|---|---|
| | R-1 ↑ | R-L ↑ | R-1 ↑ | R-L ↑ | R-1 ↑ | R-L ↑ |
| *Non-Tuned Methods* | | | | | | |
| **Non-Personalized** | 0.030 | 0.029 | 0.145 | 0.118 | 0.126 | 0.123 |
| **PAG** | 0.098 | 0.082 | 0.149 | 0.121 | 0.135 | 0.124 |
| **RAG (k=1)** | 0.101 | 0.085 | 0.152 | 0.122 | 0.149 | 0.140 |
| **RAG (k=2)** | 0.106 | 0.088 | 0.167 | 0.132 | 0.136 | 0.130 |
| **RAG (k=4)** | 0.110 | 0.092 | 0.169 | 0.135 | 0.164 | 0.157 |
| **StyleVector** | 0.104 | 0.086 | 0.156 | 0.125 | 0.132 | 0.127 |
| *Tuned Methods* | | | | | | |
| **LoRA-C** | 0.186 | 0.167 | 0.476 | 0.415 | 0.527 | 0.474 |
| **LoRA-P** | 0.120 | 0.108 | 0.489 | 0.435 | 0.398 | 0.333 |
| **LoFiT** | 0.199 | 0.179 | TLE | TLE | 0.272 | 0.256 |
| **OPPU** | 0.191 | 0.171 | 0.519 | 0.442 | **0.539** | **0.483** |
| **Ours (PerFit)** | **0.207** | **0.186** | **0.521** | **0.451** | 0.525 | 0.472 |
| *- Param. Percentage ↓ (%)* | 0.0117 | 0.0015 | 0.0039 | 0.0010 | 0.0078 | 0.0039 |
| *- Param. Reduction ↑ (%)* | 87.50 | 97.66 | 95.83 | 98.44 | 91.67 | 93.75 |

**Personalized Classification Tasks.** On classification tasks, `PerFit` achieves superior performance across all metrics. For LaMP-2N, our method attains the highest accuracy of 81.8%, surpassing OPPU by 0.8 percentage points. In LaMP-2M, `PerFit` achieves the best results with 63.0% accuracy and 51.8% F1 score, demonstrating substantial improvements over baselines. For LaMP-3, `PerFit` achieves comparable performance to OPPU with an MAE of 0.179 and RMSE of 0.443, while utilizing significantly fewer parameters.

**Personalized Generation Tasks.** In generation tasks, `PerFit` demonstrates consistent improvements over existing approaches. For LaMP-4, our method achieves the highest ROUGE-1 score

---

[2]TLE (Time Limit Exceeded) indicates that the method ran much longer than other methods on some large datasets, making it impractical to report the results in a reasonable time frame.

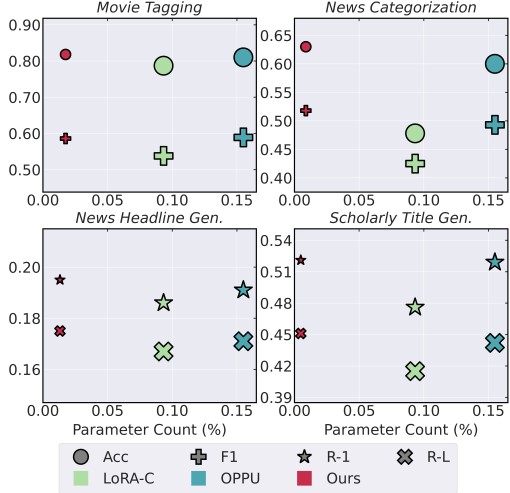
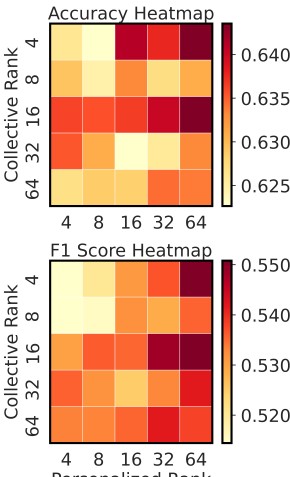

Figure 4: Performance versus parameter count on four datasets. Marker size reflects the relative training time[3].

Figure 5: Impact of Collective and Personalized Rank on Movie Tagging Performance.

of 20.7% and ROUGE-L score of 18.6%, outperforming both *Non-tuned* and *Tuned* baselines. On LaMP-5, `PerFit` achieves the best performance with a ROUGE-1 score of 52.1% and ROUGE-L score of 45.1%. While OPPU achieves marginally better performance on LaMP-7, our method maintains competitive results while utilizing significantly fewer parameters.

## 5.3 EFFICIENCY ANALYSIS (RQ2)

We conduct a detailed analysis of parameter efficiency based on the results in the main tables. As shown in Tables 2 and 3, our `PerFit` method consistently achieves state-of-the-art or highly competitive performance while dramatically reducing the number of trainable parameters. Specifically, in the first stage, `PerFit` requires only 0.0058% to 0.0117% of the total model parameters for classification tasks, and 0.0039% to 0.0117% for generation tasks. In the second stage, it uses an even smaller proportion of 0.0010% to 0.0015% for classification tasks and 0.0010% to 0.0039% for generation tasks. This two-stage design achieves a remarkable parameter reduction of **81.25%** to **98.44%** compared to strong baselines such as OPPU. This substantial reduction highlights the efficiency of our approach in both memory and computational cost. The accompanying Figure 4 provides a visual summary of these findings, plotting model performance against the proportion of trainable parameters for four representative datasets. Notably, `PerFit` not only reduces parameter count but also achieves a **17.0%** to **35.8%** reduction in training time compared to existing fine-tuning baselines. This demonstrates that our method achieves parameter and runtime efficiency without sacrificing performance, offering a practical and scalable solution for personalized LLM adaptation.

## 5.4 ABLATION STUDY (RQ3)

To validate our two-stage design and low-rank subspace intervention, we conduct an ablation study across diverse tasks (Table 4). Using only Stage-1 (collective shift learning) results in 2.6%-16.4% accuracy drops, confirming the importance of personalized adaptation in Stage-2. When training only Stage-2, both Ours@Stage-2 (C+P) and Ours@Stage-2 (P) configurations show limited performance without Stage-1's collective information. However, the higher rank configuration (C+P) still outperforms Ours@Stage-2 (P), demonstrating that increased rank helps capture more dimensions of user-specific information, though with diminishing returns. This aligns with Observation 1, suggesting that essential personalized information lies within a lower rank subspace.

---

[3]Larger markers indicate longer training times. Note that these training times refer to the first stage of training and are provided for reference only, as they are influenced by various factors including dataset size and hardware specifications. The size primarily serve to illustrate the relative time relationships between different methods.

[3]@*Stage-2* degenerates into a one-stage model, equivalent to the ReFT model detailed in Appendix D.3

Table 4: Ablation results across diverse tasks, evaluating the impact of different training stages and configurations. Here, **C** refers to the collective (Stage-1) rank, and **P** refers to the personalized (Stage-2) rank, as defined in Table 9 (*LRank* and *ULRank*, respectively). **Ours@Stage-2 (C+P)** denotes the configuration where the rank is set to the sum of both stages' ranks (*LRank + ULRank*), while **Ours@Stage-2 (P)** uses only the personalized rank (*ULRank*). *ref. LoRA-P* represents the reference values using LoRA-P.

| Method | News Categorization | | Movie Tagging | | News Headline | | Tweet Paraphrasing | |
|---|---|---|---|---|---|---|---|---|
| | Acc ↑ | F1 ↑ | Acc ↑ | F1 ↑ | R-1 ↑ | R-L ↑ | R-1 ↑ | R-L ↑ |
| **Ours** | 0.818 | 0.586 | 0.630 | 0.518 | 0.207 | 0.186 | 0.525 | 0.472 |
| @Stage-1 | 0.792 | 0.529 | 0.466 | 0.415 | 0.189 | 0.169 | 0.493 | 0.450 |
| @Stage-2 (C+P) | 0.803 | 0.604 | 0.620 | 0.496 | 0.194 | 0.175 | 0.483 | 0.438 |
| @Stage-2 (P) | 0.801 | 0.594 | 0.599 | 0.473 | 0.190 | 0.171 | 0.478 | 0.433 |
| *ref. LoRA-P* | 0.591 | 0.397 | 0.528 | 0.383 | 0.120 | 0.108 | 0.398 | 0.333 |

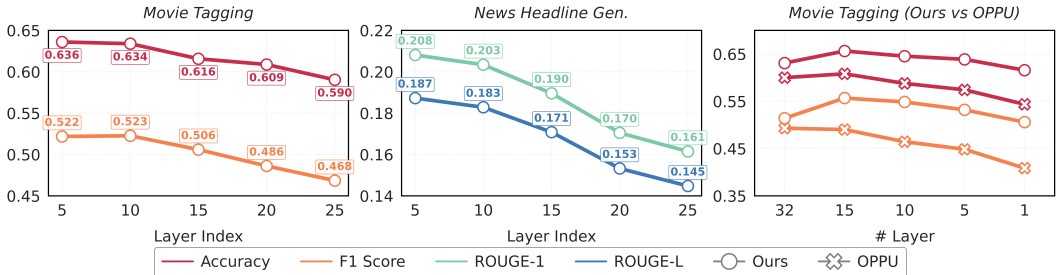

Figure 6: Layer-wise and cumulative intervention analysis. **Left & Middle:** Performance metrics (Acc, F1, R-1, R-L) versus single intervention layer position for Movie Tagging and News Headline Generation tasks. **Right:** Performance on Movie Tagging versus number of intervention layers[4].

## 5.5 HYPERPARAMETER ANALYSIS

**Layer-wise Intervention.** Figure 6 (left and middle) presents the results of intervening at a single layer for both *Movie Tagging* and *News Headline Gen.* tasks. We observe a clear trend: as the intervention layer moves from lower (earlier) to higher (later) layers, the overall performance—across all metrics—steadily declines. This finding is particularly intriguing when contrasted with prior work in knowledge editing (KE), where middle layers are typically used for learning and storing new knowledge (Meng et al., 2022). In our case, however, intervening at earlier layers yields better results. We hypothesize that this difference arises because our method's first stage must absorb and encode a large amount of user-specific information.

**Cumulative Intervention.** As shown in the right panel of Figure 6, increasing the number of intervention layers generally leads to improved performance on the *Movie Tagging* task. This suggests that leveraging more layers allows the model to better capture and utilize personalized information. However, we observe that when interventions are applied to as many as all layers, performance unexpectedly drops. This indicates that editing too many layers may introduce negative side effects, possibly due to interference or redundancy among the interventions at different layers.

**Collective vs. Personalized Rank.** Figure 5 presents a heatmap analysis of the impact of collective (Stage-1) and personalized (Stage-2) rank on *Movie Tagging* performance, measured by both accuracy and F1 score. Overall, we observe that increasing either the collective rank or the personalized rank generally leads to improved performance. However, the effect of the personalized rank appears to be more pronounced: even when the collective rank is low, a sufficiently high personalized rank can achieve near-optimal results.

---

[4]The layers are selected symmetrically around layer 15, with the spacing between layers determined as (# Model Layers / # Intervened Layer).

## 6 CONCLUSIONS AND FUTURE WORK

By uncovering fundamental patterns in user-specific information—including shared collective and unique personalized shifts—our work introduces a novel two-stage method that fine-tunes interventions directly in the hidden representation space. Extensive experiments across six diverse tasks show that this approach achieves efficient personalization with significantly reduced parameter overhead while maintaining strong performance. This work paves the way for scalable, effective personalization in intelligent systems and reveals insights into user-specific information in LLMs. Future work could explore finer-grained personalization styles, such as community-level and group-level relationships, and investigate applications in more diverse scenarios like personalized memory.

## ACKNOWLEDGMENT

The research presented in this paper was partially supported by the Research Grants Council of the Hong Kong Special Administrative Region, China (CUHK 2410072, RGC R1015-23), (CUHK 2300246, RGC C1043-24G) and CUHK 7010870.

## ETHICS STATEMENT

Personalized Large Language Models (LLMs) hold great promise for enhancing human-computer interaction and information access by enabling tailored communication and adaptive learning experiences. Their deployment also faces ethical and societal considerations. Personalization may inadvertently encode sensitive user information, posing privacy and security risks. It can also amplify biases in training data, reinforce filter bubbles or echo chambers, and increase susceptibility to personalized misinformation and manipulation, thereby threatening user autonomy. To mitigate these risks, we emphasize the need for robust privacy-preserving mechanisms, systematic bias detection and mitigation, and transparent user controls. Establishing clear ethical guidelines and fostering collaboration among researchers, ethicists, and policymakers will be essential to ensure that personalized LLMs are developed and applied responsibly, maximizing societal benefits while minimizing potential harms.

## REPRODUCIBILITY STATEMENT

To ensure the reproducibility of our work, we provide comprehensive details across multiple components of our submission. An overview of our experimental setup is presented in Section 5.1, while a detailed description is provided in Appendix D, including dataset descriptions in Appendix D.1, baseline configurations in Appendix D.3, and argument specifics in Appendix D.5. Specifically, dataset information and preprocessing steps are detailed in Appendix D.1. We use standard benchmark datasets OPPU with publicly available data splits following established protocols (Tan et al., 2024b). Source code implementing our method is made publicly available through our anonymous repository link provided in the abstract.

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

APPENDIX CONTENTS

# A    RELATED WORK

Large language models (LLMs) demonstrate remarkable abilities in text generation and complex reasoning (Radford et al.; Chang et al., 2024; Hu et al., 2024; Zhang et al., 2024d;c; Zhu et al., 2024; Wang et al., 2023; 2024a; Li et al., 2025a;b; Zhao et al., 2025e; Chen et al., 2026), thanks to comprehensive pre-training on diverse and large-scale datasets that equip them with broad general knowledge.

Recent PLLM studies cover a broad range of personalization paradigms (Huang et al., 2026; Qiu et al., 2025a; Zhao et al., 2025d; Jiang et al., 2025; Zhao et al., 2025c; Choi et al., 2025; Zhao et al., 2025a). The methods for personalized large language models (PLLMs) can be mainly divided into two types based on whether fine-tuning is involved (Liu et al., 2025; Xu et al., 2025): one type is the method that does not require fine-tuning of the large language models (LLMs), and the other type is the method that requires fine-tuning.

**Tune-Free Methods.**    Tune-free methods primarily use three approaches: input prompting, vector steering, and logits steering. (1) For input prompting, key approaches include profile-augmented generation (PAG) (Richardson et al., 2023; Qiu et al., 2025b), which uses an instruction-tuned language model to create a textual user profile from the user's personalized data, and retrieval-augmented generation (RAG) (Salemi et al., 2024a), which enhances responses by retrieving relevant entries from user history. (2) Vector steering, as implemented in StyleVector (Zhang et al., 2025), uses a separate LLM to generate contrastive pairs of personalized and non-personalized responses to modify model behavior. This method depends on pre-constructed contrastive pairs and doesn't tune model parameters; it has limited understanding of personalization. (3) Logits steering: CoS (He et al., 2024) achieves personalization by summing the logits from two rounds of outputs from the LLM: one round uses a standard prompt, while the other incorporates a user's explicit context in the prompt. Its main focus differs from our implicit personalization tasks. Related activation-editing and embedding-exploration techniques are also relevant for controllable generation (Zhao et al., 2025b; Zhu et al., 2025).

*Limitations:* Although tune-free methods are efficient because they use external data sources, their personalization capabilities are limited since they rely on historical information instead of adapting the model's internal parameters, especially for capturing users' implicit tastes and style.

**Fine-Tuning Methods.**    The one PEFT per user paradigm trains a Parameter-Efficient Fine-Tuning (PEFT) model tailored to each user using low-rank adaptation (LoRA)(Hu et al., 2021; Yang et al., 2024a;b; Zhang et al., 2024a). OPPU(Tan et al., 2024b) encodes personalized user information in PEFT parameters, enhancing the overall user experience. While research following OPPU primarily focuses on framework enhancements, such as parameter collaboration in privacy-sensitive contexts (Qi et al., 2024; Wagner et al., 2024), the area of enhancing personalized fine-tuning remains underexplored.

*Limitations.* Despite the strong performance of the LoRA architecture, it still requires millions of parameters, which poses a significant burden in personalized scenarios with a large number of users.

Note that some methods for aligning human preferences in LLMs use reinforcement learning. While these approaches vary—some requiring fine-tuning (Rame et al., 2024; Lau et al., 2024; Poddar et al., 2024; Shi et al., 2024) and others not (Chen et al., 2024) — they mainly rely on reward models based on average annotator preferences. This approach requires predefined (explicit) preferences and fails to account for how different users might want different outputs for the same prompt, which differs from our task and objectives that propose a novel personalized fine-tuning method that captures users' implicit tastes and strikes a balance between effectiveness and efficiency.

Table 5: Minimum feature dimensions $r$ needed to explain variance levels of 0.8, 0.85, 0.9, and 0.95. The ratio refers to the proportion of the feature dimension to the hidden representation dimension (4096 for Llama).

| | LaMP-2M | | LaMP-2N | | LaMP-3 | | LaMP-4 | | LaMP-5 | | LaMP-7 | |
|---|---|---|---|---|---|---|---|---|---|---|---|---|
| | $r$ | Ratio | $r$ | Ratio | $r$ | Ratio | $r$ | Ratio | $r$ | Ratio | $r$ | Ratio |
| **0.8** | 1 | 0.00121 | 1 | 0.00365 | 3 | 0.00073 | 34 | 0.02203 | 4 | 0.00098 | 3 | 0.00073 |
| **0.85** | 1 | 0.00121 | 2 | 0.00730 | 7 | 0.00171 | 91 | 0.05898 | 14 | 0.00342 | 10 | 0.00244 |
| **0.9** | 3 | 0.00362 | 4 | 0.01460 | 18 | 0.00439 | 167 | 0.10823 | 40 | 0.00977 | 22 | 0.00781 |
| **0.95** | 12 | 0.01448 | 20 | 0.07299 | 93 | 0.02271 | 368 | 0.23850 | 203 | 0.04956 | 177 | 0.04321 |

## B  SUPPLEMENTARY MATERIALS FOR THE ANALYTICAL STUDY

In this section, we provide supplementary materials related to the analytical study, including the study setup in Appendix B.1 and supplementary tables and figures in Appendix B.2. Moreover, we include further analysis to support our observations, answering the following questions:

1. Do the observations also hold in other base LLM models (Appendix B.2)?
2. Are the observations genuinely driven by personalized signals (Appendix B.3)?
3. Do the personalized shifts of the $\delta$-vectors encompass personalization (Appendix B.4)?
4. Do low-rank patterns reveal layer-wise structures? (Appendix B.5)
5. Do collective shifts stay consistent across different tasks (Appendix B.6)?

### B.1  ANALYTICAL STUDY SETUP

Using Llama2-7B as the base (i.e., non-personalized) LLM (Tan et al., 2024b;a; Kong et al., 2024), we conducted an analytical study on the widely-used personalization benchmark LaMP (Salemi et al., 2024b). To isolate the personalized information, We focus on the residual stream representation of the last token, $\mathbf{h}^\ell := \mathbf{h}_n^\ell$, which aggregates information from the entire input sequence at layer $\ell$, specifically analyzing the 16th layer following previous activation steering approaches (Arditi et al., 2025). The personalized information we concatenate for each user $\mathcal{Q}_i^{\text{orig}}$, is derived via the BM25 algorithm to identify the most relevant details of each query from the user's historical documents.

Here, we need to clarify the difference between our analysis and StyleVector (Zhang et al., 2025). StyleVector uses GPT-3.5-turbo outputs as general responses, creating contrastive pairs by combining the same query with these general outputs and the personalized outputs. However, the tokens that truly drive personalization **reside in the tokens of the query** for fine-tuning, not the last token of the response, since the query's final token influences the generation process starting from earlier tokens. Besides, relying on GPT-3.5-turbo outputs as general responses may not consistently provide a reliable baseline across different models. Moreover, the Stylevector analyzes individual user representations, like other activation steering methods, focusing on a single user dimension **without considering all users and their interrelations.**

> This analysis aims to uncover how personalized signals are represented in the hidden representation space *before response generation for all users*, so as to design the fine-tuning process directly based on the observations.

### B.2  SUPPLEMENTARY RESULTS

The supplementary figure of Figure 2 is shown in Figure 7, the figure of Figure 1 is supplemented by Table 5. This indicates that the observed pattern (Observation 2 and Observation 1) is universal across all datasets. Next, we will conduct the same analytical experiments on the Qwen model to determine whether our observations still apply to other models Figure 8. The results indicate that similar patterns persist, with the Qwen-2.5-7B model demonstrating a relatively larger shift.

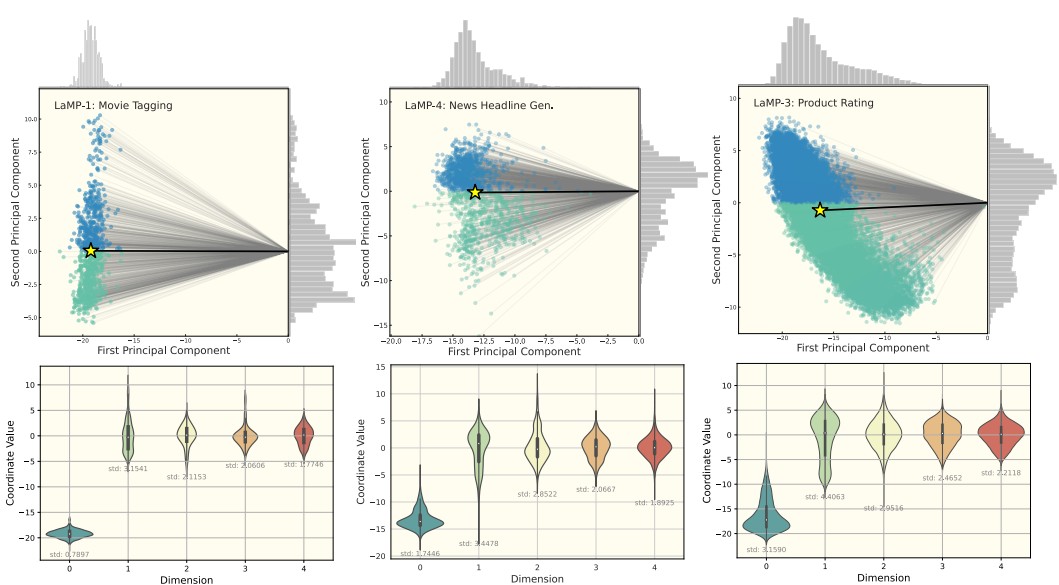

Figure 7: Supplementary figure of Figure 2 of other datasets. (1) The first row depicts the low-rank vector representations projected onto the first two principal components, with mean vectors indicated by yellow stars, demonstrating directional bias in the reduced-dimensional space. (2) The second row comprises violin plots of the coordinate value distributions across the first five feature dimensions. Here, the zeroth dimension shows a significant mean shift from zero, indicating a shared directional bias (i.e., collective shift) among users, whereas the remaining dimensions have means near zero with relatively large standard deviations, reflecting individual user variability (i.e., personalized shifts).

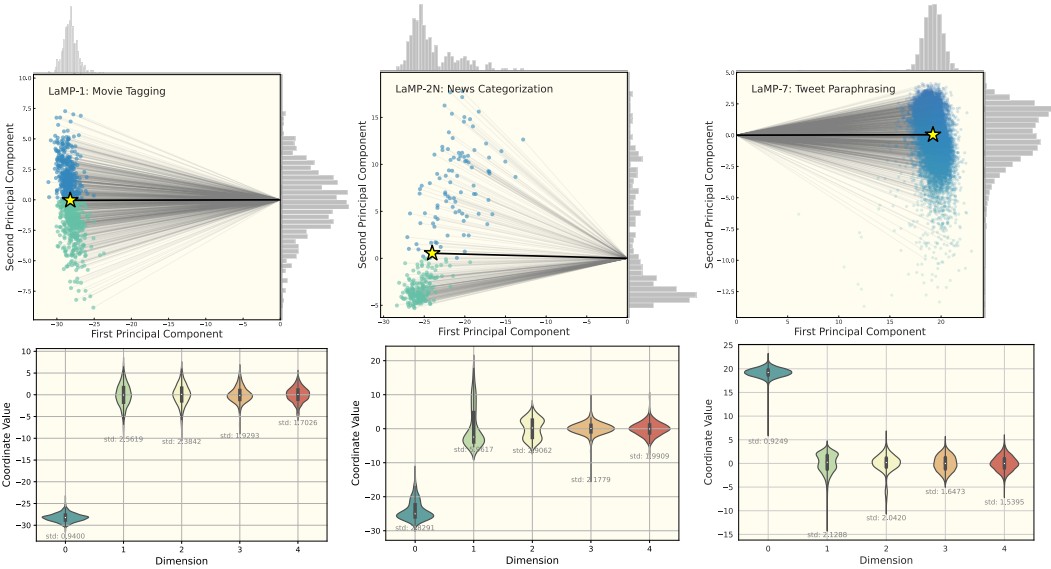

Figure 8: Projected vectors from Qwen-2.5-7B. A similar pattern also appears in the Qwen model.

Additionally, the direction of the collective shift in the dataset is consistent: the Tweet Paraphrase task tends to orient towards the positive direction, whereas other tasks show a negative orientation.

## B.3 VALIDATING PERSONALIZED SIGNALS: BEYOND INPUT LENGTH EFFECTS

As demonstrated in Appendix B.1, we introduced personalized information prior to the query to effectively isolate personalized information. In this case, the lengths of personalized input tokens are greater than those of their non-personalized counterparts. This raises a question: *"Are the observations*

**Personalized Information:**
abstract: `<abstract>`. title:
`<title>`

**Query:**
Given this author's previous publications, try to describe a template for their titles. I want to be able to accurately predict the title of one of the papers from the abstract. abstract: `<query_abstract>`. title:

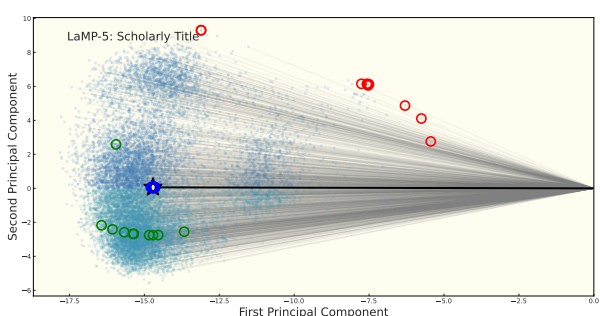

Figure 10: Personalized information template. Replace the content inside the <> with the actual descriptions of the abstract and title for each query.

Figure 11: Selected samples from the representation space. The red circle, green circle, and blue circle represent the 10 points that are farthest, at an intermediate distance, and closest to the collective vector, respectively.

*based on personalized signals instead of merely artifacts of input text length?"* Our goal in this section is to show that the phenomena observed are attributed to personalized representations.

To substantiate this claim, we conducted an ablation study in which the personalized information prepended to each query was systematically substituted with randomly generated text of equivalent length. This careful and controlled manipulation enabled us to closely examine the distribution of $\delta$-vectors within the representation space, thereby providing valuable insights into the role of personalized signals in shaping our observations, while effectively eliminating the influence of other interfering factors. The results show Figure 9 that the random prefix does cause a slight shift in the vector representation; however, the degree of this shift is much smaller than the significant deviation caused by $\delta$-vectors . This indicates that our observations are indeed based on personalized information rather than merely the added text length of the prefix to the query.

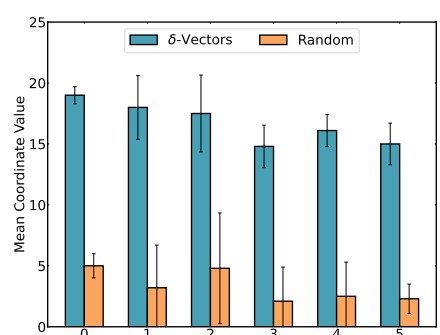

Figure 9: Comparison of the mean absolute coordinate values between $\delta$-vectors and random vectors.

### B.4 PERSONALIZED SHIFTS: A CASE STUDY

This section aims to answer the question: "*Do the personalized shifts of the $\delta$-vectors encompass personalization?*" The personalized information regarding implicit styles in the LaMP dataset is challenging to quantify. To explore this, we select samples within the representation space for a case study to determine whether nearby representations exhibit similar styles.

For instance, in the context of Tweet Paraphrase, the template of added personalized information and the queries is illustrated in Figure 10. Using the collective shift vector as a reference, we identify the top 10 users with the nearest vectors, the top 10 vectors at an intermediate distance, and the 10 farthest vectors. We then present their corresponding personalized information regarding the titles generated by the user previously, aligning with the user's preferences, as illustrated in Figure 12, Figure 13, and Figure 14, respectively. The points we select are highlighted in Figure 11.

From these examples, it is evident that while the style may not convey significant information, users with personalized vectors that represent different regions in the embedding space exhibit clear and intuitive differences in their corresponding personalized information.

Moreover, an interesting discovery is that the **points that are farther from the collective shift tend to be outliers.** The queries associated with these outlier points lack effective information, resulting in the inability to incorporate meaningful personalization. Consequently, compared to the

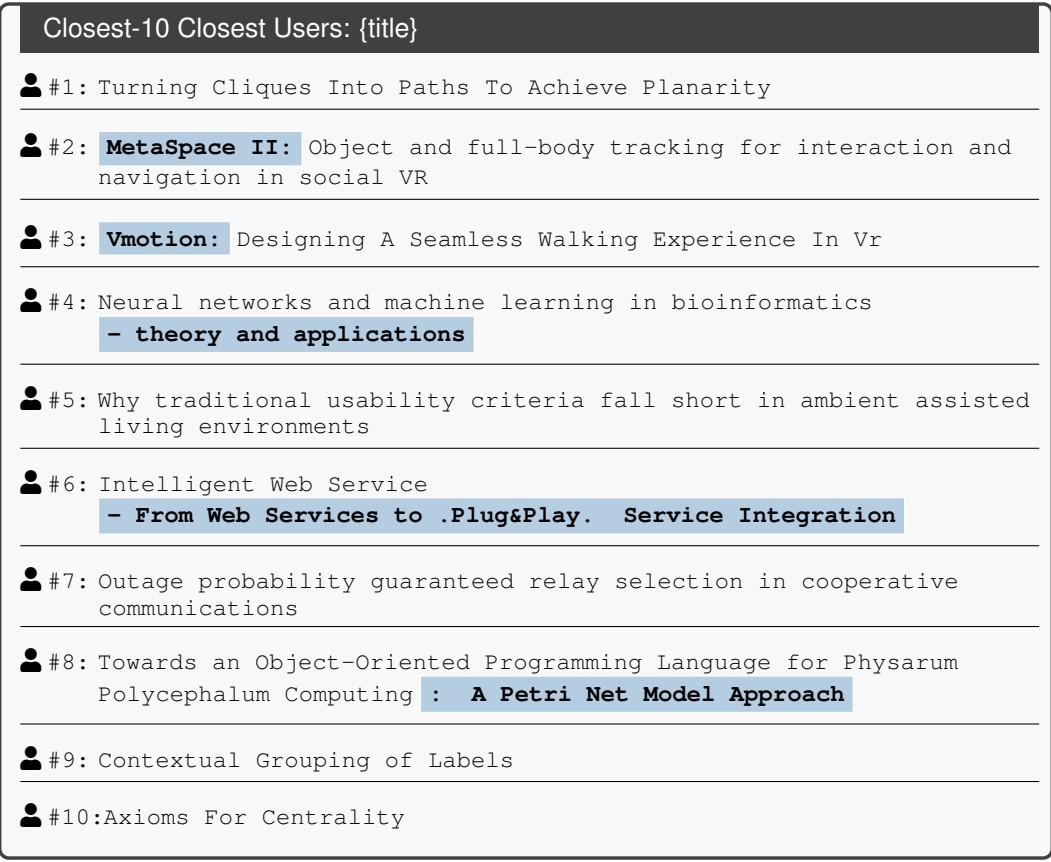

Figure 12: Closest-10 closest users. These sentences indicate that a majority of individuals prefer to employ punctuation marks, such as commas and dashes.

collective shift, there is a considerable deviation in these instances. This further validates and supports the direct correlation between $\delta$-vectors and personalized information in the embedding space. It also demonstrates that the personalized shift, based on the collective shift, can effectively reflect individualized information.

### B.5 Low-Rank Patterns: A Layer-Wise Analysis

Building on the previous findings (Observation 1) that the $\delta$-vectors in the middle layers display a pronounced low-rank structure, this section extends the investigation to address the question: *"Does the low-rank property consistently appear at every layer of the model?"* To address this question, the layer-wise ranks of $\delta$-vectors were computed for each dataset, as shown in Figure 15. Two key observations emerge from this analysis.

First, the results of the first two subfigures of Figure 15 reveal a consistent trend: **early and middle layers exhibit relatively low ranks**, while a sharp increase emerges after layer 15, particularly for generation tasks. This pattern aligns with prior findings showing that deeper layers of large language models increasingly specialize in task-level semantics and generation (Geva et al., 2021; Ji et al., 2024; Song et al., 2025), leading to heavily entangled user- and task-related features. Consequently, these layers are less suitable for isolating and controlling style-level personalization. In contrast, top–middle layers compress user inputs into shared semantic representations (Wang et al., 2024c), thereby making stylistic shifts more cleanly expressed and naturally forming a low-rank personalization subspace.

Second, further evidence comes from projecting user representations onto the first SVD component (right panel in Figure 15). Early and middle layers show **minimal variance across users, reflecting a strong collective shift.** In contrast, deeper layers exhibit substantially larger variance,

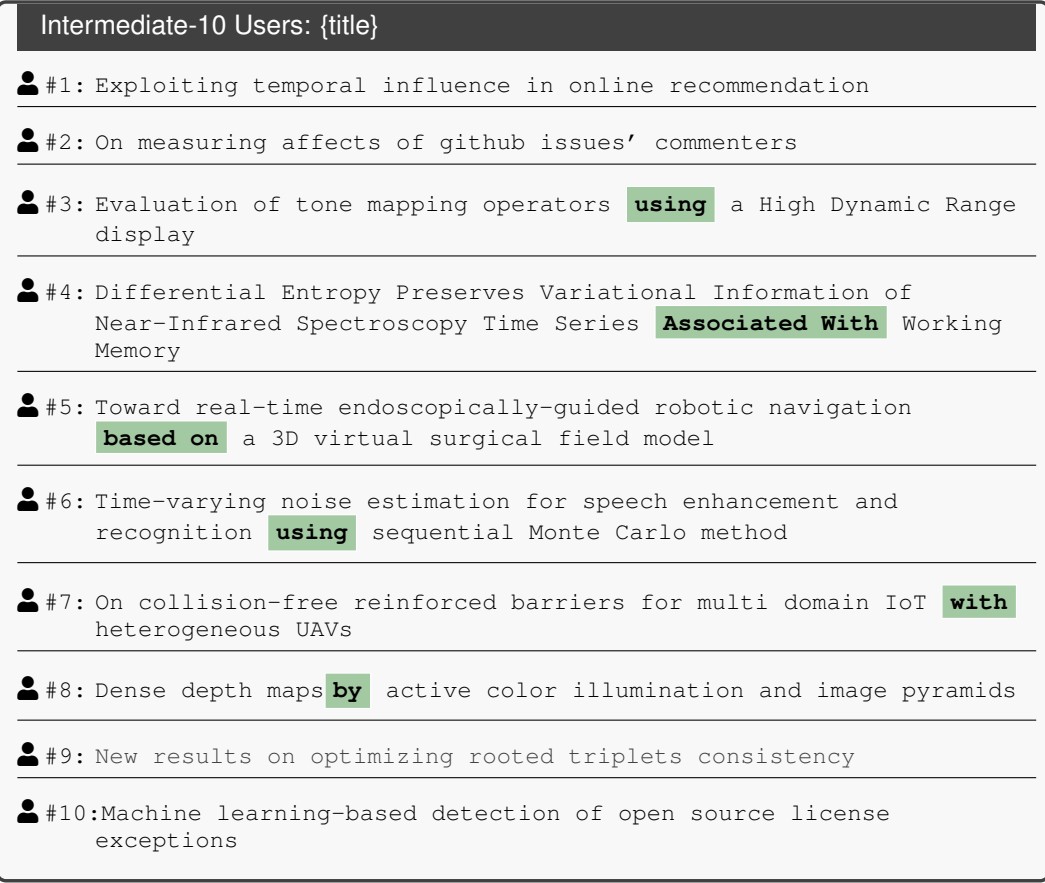

Figure 13: Intermediate-10 closest users. The titles of these examples rarely use punctuation marks; instead, they favor terms such as 'based on' and 'using' to indicate specific methodologies. Additionally, the descriptions of the titles are more precise compared to those of the closest users.

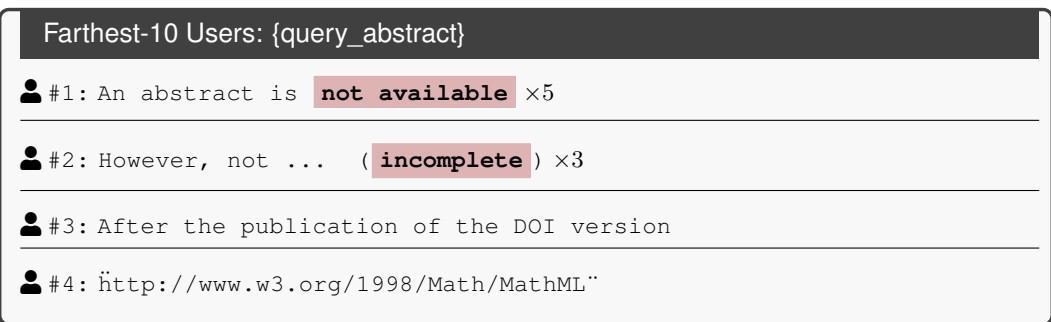

Figure 14: Farthest-10 closest users. The distant points are all outliers, and the query of users lacks effective abstract information. As shown above, there is no indication of the user's methodology.

indicating that user representations become more dispersed as stylistic and task-specific factors intertwine—consistent with the increased entanglement observed in later layers.

It is worth noting that although the $\delta$-vectors at the final layer also appear low-rank, their representation directions are highly dispersed, showing no collective shift and large variance in user-specific offsets. Thus, it is unsuitable for intervention. This distinctive behavior of the last layer has also been highlighted in prior work (Skean et al., 2024).

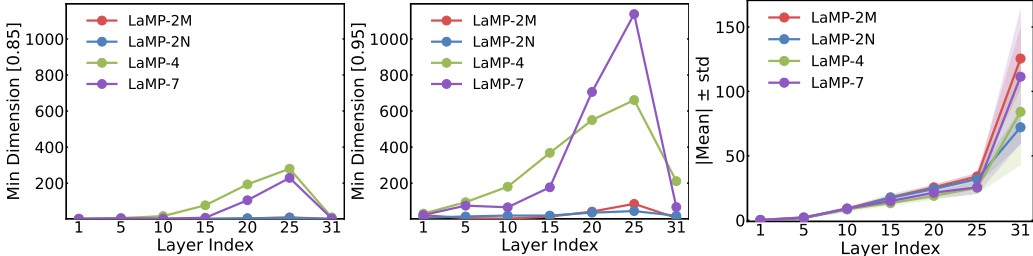

Figure 15: Layer-wise analysis of Observation 1. The first two panels show, for each layer, the minimum feature dimension $r$ needed to explain variance levels of $0.85$ and $0.95$, revealing the layer-wise rank patterns. The right panel reports $|\text{mean}| \pm \text{std}$ of users' coordinates on the first principal component after SVD; for each task and layer, the mean is computed over all users.

The observations in this section are consistent with the experimental analysis in Appendix E.2, that interventions applied around layers 5–10 yield the strongest performance, while those at deeper layers interfere degrade personalization quality, especially at the last layer.

In summary, the low-rank and collective shift personalized patterns (Observation 1 and Observation 2) reside mainly in the early–middle layers, not in the deep layers, where they entangled generative factors and distort personalization signals.

### B.6  TASK-SPECIFIC PATTERNS: A CROSS-TASK ANALYSIS

The previous analysis demonstrated that personalized information exhibits a collective shift. We further explored the relationships among the personalized vectors in the LaMP dataset, as illustrated in Figure 16. Our findings indicate that the collective shifts differ across various tasks. For instance, classification tasks, such as Movie Tagging and News Categorization, show a closer distribution compared to text generation tasks. Specifically, tasks like News Headline generation and Scholarly Title generation exhibit a more similar pattern. The distribution of Tweet Paraphrase differs from that of other personalization tasks, which aligns with our findings regarding the variation in their shift directions in Figure 2. These insights suggest that when developing personalized LLMs in the future, **it is important to consider inter-task relationships**; especially in complex tasks and scenarios, effective analysis and explanations can be derived from their representations. This type of analysis can enhance our understanding of the underlying dynamics within personalized models.

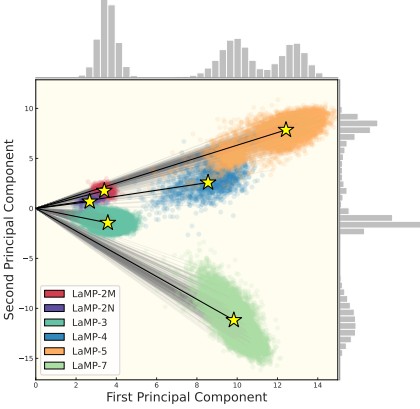

Figure 16: Representation of combined $\delta$-vectors from diverse datasets, with each color denoting a distinct dataset.

To summarize, this section validates the following key observations and conclusions:

- The observations are universal across multiple datasets, tasks, and LLMs.
- $\delta$-vectors reflect personalized information, confirmed by isolating interfering factors.
- Personalized shifts in the representation space correlate with users' stylistic traits.
- Personalized patterns primarily reside in the early–middle layers.
- Differences in collective shift may exist across tasks.

## C SUPPLEMENTARY MATERIALS FOR METHODOLOGY

### C.1 ALGORITHM PSEUDOCODE OF PERFIT

**Note that**, for simplicity in implementation, we do not initialize the Stage 2 parameters $\Delta\Theta^{(2)}$ separately for each user; instead, they are computed directly during Stage 1 initialization. This is not the main focus or core contribution of our work and can be flexibly modified if needed. Here, we present the details of this procedure in the pseudocode (Algorithm 1).

---

**Algorithm 1:** PerFit

**Input** : Users $\mathcal{U} = \{u_i\}_{i=1}^{N}$, queries $\mathcal{Q}_i$, targets $\mathcal{Y}_i$ for each $u_i$, base (non-personalized) LLM $M_0$ with parameters $\Theta_0$   // Section 2.1

**Output :** Personalized models $\{M_i = \Theta_0 + (\Delta\Theta^{(1)}, \Delta\Theta_i^{(2)})\}_{i=1}^{N}$ for each user $u_i \in \mathcal{U}$

1 Initialize parameters of Stage 1 and Stage 2: $\Delta\Theta = \{\Delta\Theta^{(s)}\}_{s=1}^{2} = \{(\mathbf{R}^{(s)}, \mathbf{W}^{(s)}, \mathbf{b}^{(s)})\}_{s=1}^{2}$.

2 **Function** Phi_stage $(\mathbf{x}, \Delta\Theta^{(s)})$:

3 $\quad$ **return** $\mathbf{x} + \mathbf{R}^{(s)\top}(\mathbf{W}^{(s)}\mathbf{x} + \mathbf{b}^{(s)} - \mathbf{R}^{(s)}\mathbf{x})$ ; // Equation (2)

4 **Stage 1: Train shared intervention** ;

5 **for** *each user $u_i \in \mathcal{U}$* **do**

6 $\quad$ **for** *each query $q_j^{(i)} \in \mathcal{Q}_i$ with target $y_j^{(i)} \in \mathcal{Y}_i$* **do**

7 $\quad\quad$ Compute $\mathbf{h}_j$ from $M_0(q_j^{(i)})$;

8 $\quad\quad$ Compute intervened representation $\tilde{\mathbf{h}}_j \leftarrow$ Phi_stage $(\mathbf{h}_j, \Delta\Theta^{(1)})$;

9 $\quad\quad$ Compute intervened representation $\tilde{\mathbf{h}}_j \leftarrow$ Phi_stage $(\tilde{\mathbf{h}}_j, \Delta\Theta^{(2)})$; // Equation (1)

10 $\quad\quad$ Compute loss $\mathcal{L}(\tilde{y}_j, y_j^{(i)})$ where $\tilde{y}_j$ decoded from $\tilde{\mathbf{h}}_j$;

11 $\quad$ **end**

12 **end**

13 Update $\Delta\Theta^{(1)}$ and $\Delta\Theta^{(2)}$ to minimize total loss over all users;

14 **Stage 2: Fine-Tune user-specific interventions** ;

15 **for** *each user $u_i \in \mathcal{U}$* **do**

16 $\quad$ $\Delta\Theta_i^{(2)} \leftarrow \Delta\Theta^{(2)}$ ;

17 $\quad$ **for** *each query $q_j^{(i)} \in \mathcal{Q}_i$ with target $y_j^{(i)} \in \mathcal{Y}_i$* **do**

18 $\quad\quad$ Compute $\mathbf{h}_j$ from $M_0(q_j^{(i)})$;

19 $\quad\quad$ Compute personalized representation $\hat{\mathbf{h}}_j \leftarrow$ Phi_stage(Phi_stage $(\mathbf{h}_j, \Delta\Theta^{(1)})$, $\Delta\Theta_i^{(2)}$);

20 $\quad\quad$ Compute loss $\mathcal{L}(\hat{y}_j, y_j^{(i)})$ where $\hat{y}_j$ decoded from $\hat{\mathbf{h}}_j$;

21 $\quad$ **end**

22 $\quad$ Update $\Delta\Theta_i^{(2)}$ to minimize total loss for user $u_i$;

23 **end**

24 **return** personalized models $\{M_i = \Theta_0 + (\Delta\Theta^{(1)}, \Delta\Theta_i^{(2)})\}_{i=1}^{N}$;

---

This algorithm takes a set of users with their queries and targets, and a base model $M_0$ with parameters $\Theta_0$. Line 1 initializes shared Stage 1 and Stage 2 intervention parameters.

- In Stage 1 (lines 4–13), for each user and query, the base model encodes the query, then the representation is transformed successively by shared Stage 1 and Stage 2 functions, and the resulting output from the base model after intervention is used to compute loss for updating both shared parameter sets.

- In Stage 2 (lines 14–23), user-specific Stage 2 parameters are fine-tuned individually by applying the shared Stage 1 followed by the personalized Stage 2 transformation on the encoded queries, only fine-tuning $\Delta\Theta^{(2)}$.

The algorithm returns personalized models composed of the base parameters plus the shared Stage 1 and user-specific Stage 2 parameters (line 24). This two-stage approach enables learning a shared adaptation and personalized refinements efficiently.

Besides the description above, the representation $\mathbf{h}$ can correspond to the output of any one or multiple layers from the first layer to the last layer $\mathcal{L}$ of the model; if multiple layers are used, then

each layer's representation $\mathbf{h}^{(\ell)}$ requires its own set of intervention parameters, i.e., $\Delta\Theta^{(\ell)(1)}$ and $\Delta\Theta^{(\ell)(2)}$, respectively. For simplicity, we also omit specifying the exact layer(s) here.

## C.2 PARAMETER EFFICIENCY ANALYSIS

For a weight matrix $W \in \mathbb{R}^{d_{out} \times d_{in}}$, LoRA decomposes the weight update $\Delta W = AB$ with $A \in \mathbb{R}^{d_{out} \times r}, B \in \mathbb{R}^{r \times d_{in}}$, incurring parameter count $N_{\text{LoRA}} = r(d_{in} + d_{out}) \approx 2r(d_{in} + d_{out})$. Conversely, PerFit $\phi_{\Delta\Theta}$ operates on representation $x \in \mathbb{R}^d$ via the transformation $\phi_{\Delta\Theta}(x) = x + \mathbf{R}^\top (\mathbf{W}^{(s)}x + \mathbf{b}^{(s)}) - \mathbf{R}^\top \mathbf{R}x$, where $\mathbf{R} \in \mathbb{R}^{r \times d_{out}}, \mathbf{W}^{(s)} \in \mathbb{R}^{r \times d_{out}}, \mathbf{b}^{(s)} \in \mathbb{R}^r$, with parameter count $N_{\phi_{\Delta\Theta}} \approx 2rd_{out}$; however, LoRA requires $M$ such blocks per layer, resulting in $N_{\text{LoRA, layer}} = M \times 2rd$, PerFit necessitates only one module per representation layer with $N_{\phi_{\Delta\Theta}, \text{layer}} = 2rd$, thereby achieving substantial parameter efficiency when $M \gg 1$. From the experiments in Figure 6 (right), our one-layer method outperforms the best setting of LoRA, which combines all layers with LoRA. Therefore, the total number of parameters required by LoRA is significantly smaller than that of our method.

## C.3 TWO-STAGE PEFT VS. PERFIT

Consider the hidden state update at layer $\ell$ of a decoder-style Transformer:

$$\mathbf{h}_t^{\ell+1} = \mathbf{h}_t^\ell + \text{FFN}_0^\ell\left(\mathbf{h}_t^\ell + \text{MHA}_0^\ell(\mathbf{h}_{1:t}^\ell)\right),$$

as defined in Section 2.2 (with hidden dimension $d$). We study whether LoRA-based interventions on FFN or MHA layers can induce a *representation shift* that can be expressed as an *input-independent, additive constant vector* decomposable into a collective and a personalized shift as our observations:

$$\mathbf{h} \mapsto \mathbf{h} + \mathbf{s}_c + \mathbf{s}_p.$$

**Proposition 1.** *LoRA-induced shifts $\Delta_{\text{LoRA}}(\mathbf{x})$ in FFN/MHA are input-dependent, piecewise-affine functions and cannot, except in degenerate cases, equal a constant additive vector $\mathbf{s}_c + \mathbf{s}_p$.*

*Proof.* Consider a two-layer MLP:

$$\text{FFN}(\mathbf{x}) = \mathbf{W}_2 \sigma(\mathbf{W}_1\mathbf{x} + \mathbf{b}_1) + \mathbf{b}_2, \quad \sigma = \text{ReLU}.$$

After inserting LoRA at $\mathbf{W}_1$ and $\mathbf{W}_2$, the modified network becomes

$$\text{FFN}_{\text{LoRA}}(\mathbf{x}) = (\mathbf{W}_2 + \mathbf{B}_2\mathbf{A}_2)\,\sigma\big((\mathbf{W}_1 + \mathbf{B}_1\mathbf{A}_1)\mathbf{x} + \mathbf{b}_1\big) + \mathbf{b}_2,$$

and the induced shift is

$$\Delta_{\text{LoRA}}(\mathbf{x}) = \text{FFN}_{\text{LoRA}}(\mathbf{x}) - \text{FFN}(\mathbf{x}).$$

To analyze this shift, define

$$\mathbf{z}_0 = \mathbf{W}_1\mathbf{x} + \mathbf{b}_1, \quad \mathbf{z}_1 = (\mathbf{W}_1 + \mathbf{B}_1\mathbf{A}_1)\mathbf{x} + \mathbf{b}_1, \quad \mathbf{D}_i = \text{diag}(\mathbf{1}_{\mathbf{z}_i > 0}).$$

Here $\mathbf{D}_i$ are binary diagonal matrices representing the input-dependent ReLU gates. Expanding gives

$$\Delta_{\text{LoRA}}(\mathbf{x}) = (\mathbf{B}_2\mathbf{A}_2\mathbf{D}_1 + \mathbf{W}_2(\mathbf{D}_1 - \mathbf{D}_0)\mathbf{W}_1)\mathbf{x} + (\mathbf{B}_2\mathbf{A}_2\mathbf{D}_1 + \mathbf{W}_2(\mathbf{D}_1 - \mathbf{D}_0))\mathbf{b}_1.$$

This expression shows that $\Delta_{\text{LoRA}}(\mathbf{x})$ is an affine function of $\mathbf{x}$ within each ReLU region (where $\mathbf{D}_0, \mathbf{D}_1$ are fixed). Crucially, both the effective linear term

$$\mathbf{B}_2\mathbf{A}_2\mathbf{D}_1 + \mathbf{W}_2(\mathbf{D}_1 - \mathbf{D}_0)\mathbf{W}_1$$

and the bias term depend on the gating matrices $\mathbf{D}_0, \mathbf{D}_1$, which in turn depend on $\mathbf{x}$. Therefore, across different regions of the input space, $\Delta_{\text{LoRA}}(\mathbf{x})$ changes discontinuously and is piecewise-affine.

For $\Delta_{\text{LoRA}}(\mathbf{x})$ to be a constant vector independent of $\mathbf{x}$, the effective matrix must vanish for *all* possible pairs $(\mathbf{D}_0, \mathbf{D}_1)$. This imposes extremely restrictive algebraic constraints on $(\mathbf{W}_1, \mathbf{W}_2, \mathbf{A}_1, \mathbf{B}_1)$, which only hold in measure-zero degenerate cases. Thus, in general, LoRA cannot induce an input-independent constant shift.

The situation becomes even stricter for MHA layers: the softmax introduces multiplicative query–key coupling, yielding nonlinear dependencies on $\mathbf{x}$. Hence $\Delta_{\text{LoRA}}(\mathbf{x})$ cannot represent a constant additive vector in attention either. □

**Proposition 2.** *Two-stage LoRA updates are not additive:*

$$\Delta_{\text{LoRA}}^{(p \circ c)}(\mathbf{x}) \neq \Delta_{\text{LoRA}}^{(c)}(\mathbf{x}) + \Delta_{\text{LoRA}}^{(p)}(\mathbf{x}).$$

*Proof.* Stage $p$ operates on hidden states already altered by stage $c$, so its gating matrices differ from those under raw input $\mathbf{x}$. Hence the resulting $\Delta_{\text{LoRA}}$ cannot decompose additively. □

**Proposition 3.** `PerFit` *yields approximately additive shifts in a low-rank subspace:*

$$(\phi_{\Delta\mathbf{\Theta}^{(2)}} \circ \phi_{\Delta\mathbf{\Theta}^{(1)}})(\mathbf{x}) = \mathbf{x} + \mathbf{v}^{(1)}(\mathbf{x}) + \mathbf{v}^{(2)}(\mathbf{x}) + O(\|\mathbf{D}\mathbf{v}^{(2)}\| \cdot \|\mathbf{v}^{(1)}(\mathbf{x})\|).$$

*Proof.* Let $\mathbf{v}(\mathbf{x}) = \mathbf{R}^\top(\mathbf{W}\mathbf{x} + \mathbf{b} - \mathbf{R}\mathbf{x}) \in \text{Im}(\mathbf{R}^\top)$. Then

$$(\phi_{\Delta\mathbf{\Theta}^{(2)}} \circ \phi_{\Delta\mathbf{\Theta}^{(1)}})(\mathbf{x}) = \mathbf{x} + \mathbf{v}^{(1)}(\mathbf{x}) + \mathbf{v}^{(2)}(\mathbf{x} + \mathbf{v}^{(1)}(\mathbf{x})).$$

First-order expansion:

$$\mathbf{v}^{(2)}(\mathbf{x} + \mathbf{v}^{(1)}(\mathbf{x})) = \mathbf{v}^{(2)}(\mathbf{x}) + O(\|\mathbf{D}\mathbf{v}^{(2)}\|\|\mathbf{v}^{(1)}(\mathbf{x})\|).$$

Thus both collective and personal shifts lie in $\text{Im}(\mathbf{R}^\top)$ and combine additively up to a small, controllable cross term. □

> **Remark 1.** *The above results highlight a key distinction: LoRA induces input-dependent, non-additive shifts, making it ill-suited for modeling clean personalization. This explains OPPU's performance degradation under fewer adapted layers (Table 11). In contrast,* `PerFit` *enforces low-rank additive structure, yielding interpretable and controllable collective + personal shifts, perfectly aligning our observations.*

# D    SUPPLEMENTARY MATERIALS FOR THE EXPERIMENTAL SETUP

## D.1    DATASETS

Our experiments utilize the LaMP benchmark (Salemi et al., 2024b), a collection of personalization tasks from which we select six distinct tasks - three for classification and three for generation[5]. In alignment with the OPPU framework (Tan et al., 2024b), we recognize the importance of substantial user history for effective model personalization. Consequently, we identify and select about 100 most prolific users (those with the most extensive interaction histories) from the time-ordered LaMP variant to serve as our test cohort. The remaining users' data is allocated for training the collective LLM in the first stage. The dataset statistics are listed in Table 6.

## D.2    METRICS

We employ a comprehensive set of evaluation metrics to assess model performance across different tasks. For classification tasks, we utilize Accuracy (Acc), F1 Score (F1), Mean Absolute Error (MAE), and Root Mean Squared Error (RMSE). For generation tasks, we employ ROUGE-1 (R-1) and ROUGE-L (R-L) metrics to evaluate the quality of generated text. The detailed specifications of these metrics are presented in Table 7, where $TP$, $TN$, $FP$, and $FN$ represent true positives, true negatives, false positives, and false negatives respectively; $y_i$ and $\hat{y}_i$ denote the ground truth and predicted values; $S$ and $R$ represent the generated and reference sequences; and $LCS(S, R)$ indicates the length of the longest common subsequence between $S$ and $R$.

---

[5]We omitted the LaMP-1 citation dataset because we were unable to reproduce results using the OPPU prompt, and for most queries, we could not get outputs in the required format. This may be due to limitations in Llama2's instruction following capabilities. Like OPPU, we did not use the LaMP-6 dataset due to privacy concerns. The remaining six datasets still ensure task diversity.

Table 6: Dataset statistics for the LaMP benchmark. We present the average sequence length measured in token count, where #Q represents the quantity of queries, $L_{in}$ and $L_{out}$ denote the average input and output sequence lengths respectively, #History indicates the volume of historical interactions, and #Classes shows the number of classification categories for classification tasks. #Users shows the number of users in the base LLM training stage and personal PEFT training stage (format: first stage/second stage).

| Task | #Users | Base LLM Training | | | Personal PEFT Training | | | | |
|------|--------|------|------|------|------|------|------|------|------|
| | | #Q | $L_{in}$ | $L_{out}$ | #Q | #History | $L_{in}$ | $L_{out}$ | #Classes |
| 2M | 829 / 100 | 3,181 | 92.1 | - | 3,302 | 55.6 | 92.6 | - | 15 |
| 2N | 274 / 49 | 3,662 | 68.2 | - | 6,033 | 219.9 | 63.5 | - | 15 |
| 3 | 19,899 / 101 | 22,388 | 128.7 | - | 112 | 959.8 | 211.9 | - | 5 |
| 4 | 1,543 / 100 | 7,275 | 33.9 | 9.2 | 6,275 | 270.1 | 25.2 | 11.1 | - |
| 5 | 14,581 / 101 | 16,075 | 162.1 | 9.7 | 107 | 442.9 | 171.6 | 10.3 | - |
| 7 | 13,337 / 100 | 14,826 | 29.7 | 18.3 | 109 | 121.2 | 29.4 | 18.0 | - |

Table 7: Specifications of evaluation metrics used in our experiments. For each metric, we indicate the corresponding dataset, task type, and mathematical formulation.

| Metric | Dataset | Task Type | Formulation |
|--------|---------|-----------|-------------|
| Acc | LaMP-2N, 2M | Classification | $\frac{TP+TN}{TP+TN+FP+FN}$ |
| F1 | LaMP-2N, 2M | Classification | $2 \times \frac{precision \times recall}{precision + recall}$ |
| MAE | LaMP-3 | Classification | $\frac{1}{n}\sum_{i=1}^{n}|y_i - \hat{y}_i|$ |
| RMSE | LaMP-3 | Classification | $\sqrt{\frac{1}{n}\sum_{i=1}^{n}(y_i - \hat{y}_i)^2}$ |
| R-1 | LaMP-4,5,7 | Generation | $\frac{|S \cap R|}{|R|}$ |
| R-L | LaMP-4,5,7 | Generation | $\frac{LCS(S,R)}{|R|}$ |

## D.3 BASELINES

To provide a comprehensive comparison, we evaluate our method against a diverse set of baselines, categorized into **Non-Tuned** and **Tuned** approaches. All baseline models are implemented using Llama2-7B[6] as the foundation model.

**Non-Tuned Methods**

- **Non-Personalized**: This approach utilizes the pre-trained model without any modifications to generate responses for user queries. It establishes a performance floor for our experiments and serves as a reference point for measuring the effectiveness of personalization techniques.

- **Profile Augmented Generation (PAG)**: This method synthesizes a textual user profile using an instruction-tuned language model (e.g., Vicuna-13B[7]), derived from the user's interaction history. The generated profile is then prepended to each query to provide explicit contextual information about user preferences, enabling the model to generate more personalized responses without parameter updates.

- **Retrieval Augmented Generation (RAG)**: This technique implements the BM25 algorithm to retrieve the most relevant entries from a user's history (with $k = 1, 2, 4$) for each query. These retrieved entries serve as supplementary context for the model during inference, allowing it to access specific historical interactions that may be relevant to the current query.

- **StyleVector** (Zhang et al., 2025): This framework represents a training-free approach that disentangles and encodes personalized writing style as a vector within the LLM's activation space. StyleVector enables style-controlled generation during inference without requiring retrieval mechanisms or parameter storage. The style vector is computed as the mean

---

[6]Llama2-7B open-source model: https://huggingface.co/meta-llama/Llama-2-7b-hf
[7]Vicuna-13B open-source model: https://lmsys.org/blog/2023-03-30-vicuna/

difference between positive and negative exemplars, and is injected into a specific token representation at a predetermined layer. Unlike our proposed method, StyleVector depends on carefully selected sample pairs, making it particularly sensitive to data quality and quantity, and necessitates more sophisticated vector engineering.

**Tuned Methods**

- **Personalized LoRA (LoRA-P)** (Hu et al., 2021): This standard parameter-efficient fine-tuning (PEFT) methodology creates individual models that are fine-tuned on each user's historical data. The approach produces user-specific parameter adaptations that capture individual preferences and behaviors through low-rank matrix decompositions of weight updates.

- **Collective LoRA (LoRA-C)** (Hu et al., 2021): This method employs LoRA fine-tuning on the collective history of all users excluding the 100 test users. The approach quantifies the benefits of collaborative training without personalization and provides a model that captures general user behaviors rather than individual preferences.

- **LoFiT** (Yin et al., 2024): This method identifies a subset of attention heads that are most important for learning a specific task, and then trains offset vectors to add to the model's hidden representations at those selected heads. LoFiT requires a two-step process for each stage: *attention head selection* followed by *bias tuning*. While this approach enables targeted adaptation, it introduces additional training overhead and computational cost compared to other methods.

- **OPPU** (Tan et al., 2024b): This technique integrates a two-stage approach combining collaborative and personalized fine-tuning. The first stage trains on collective user data (similar to LoRA-C), while the second stage adapts these parameters to individual users (similar to LoRA-P). This dual-stage process allows the model to benefit from both collective knowledge and individual customization.

## D.4 IMPLEMENTATION DETAILS

Key training settings are consistent across both training stages: we use the AdamW optimizer with a learning rate of $1 \times 10^{-4}$, weight decay of $1 \times 10^{-2}$, and BF16 precision. Gradient clipping is applied with a maximum norm of 0.3. Batch sizes are generally 16, with exceptions for Product Rating (batch size 2) and Scholarly Title Generation (batch size 4) due to computational requirements. The base LLM is trained for 3 epochs, and the personal PEFT stage for 2 epochs. For inference, we set the temperature to 0.1, top-k sampling to 10, and top-p sampling to 0.9. PEFT-based methods (LoRA, OPPU) utilize a LoRA rank $r = 8$ and $\alpha = 8$. For our representation-based methods, hyperparameters such as low-rank dimensions, intervention layers, and positions were determined via a 20-trial random search.

## D.5 ARGUMENTS

This section outlines the key experimental configurations used in our study, including training arguments, inference parameters, and model-specific hyperparameters for both PEFT and representation-based methods. These configurations were carefully chosen to ensure fair comparisons while optimizing performance across all approaches.

### D.5.1 TRAINING ARGUMENTS

We maintain consistent optimization parameters across both training stages, employing the AdamW optimizer with a learning rate of $1 \times 10^{-4}$, weight decay of $1 \times 10^{-2}$, and a warmup ratio of 0.1. All models are trained using Brain Floating Point 16-bit precision (BF16) to balance computational efficiency and numerical stability. We apply gradient clipping with a maximum gradient norm of 0.3 and utilize a linear learning rate scheduler. For efficient batch processing, we implement group_by_length=True to minimize padding overhead. The batch size is set to 16 for most tasks, with exceptions for Product Rating (batch size=2) and Scholarly Title (batch size=4) due to their specific computational requirements. The training epochs differ between stages: 3 epochs for the base LLM

training stage and 2 epochs for the personal PEFT training stage, unless otherwise specified for the second stage.

### D.5.2 Inference Arguments

During inference, we carefully control the generation process to ensure consistent and high-quality outputs. We set the temperature parameter to 0.1, which produces more deterministic and focused responses by reducing randomness in the token selection process. For sampling, we employ both top-k and top-p (nucleus) sampling strategies in combination: top-k is set to 10, limiting consideration to only the 10 most probable next tokens, while top-p is set to 0.9, ensuring that the model samples from tokens comprising the top 90% of the probability mass. These parameters strike a balance between output diversity and coherence, allowing for some controlled variation while maintaining response quality and relevance to the personalized context.

### D.5.3 Model Hyperparameters

For our experiments, we carefully configured the hyperparameters for both parameter-efficient fine-tuning (PEFT) and representation-based fine-tuning (ReFT) methods to ensure fair comparison.

**PEFT Methods (LoRA-P/LoRA-C/OPPU)** For PEFT methods, we maintained consistent LoRA configurations across both training stages with a rank ($r$) of 8 to control the dimensionality of LoRA matrices, a scaling factor ($\alpha$) of 8 to determine the magnitude of LoRA updates, and a dropout rate of 0.05 to regulate overfitting in LoRA layers. In the first stage (Base LLM Training), we applied these configurations to all projection matrices including "q_proj", "v_proj", and "k_proj" modules, while in the second stage (Personal PEFT Training), we focused only on the query and value projection matrices ("q_proj" and "v_proj").

**Localized Fine-Tuning (LoFiT)** Following the original LoFiT implementation (Yin et al., 2024), we conducted a two-step process for the personalization training stage: attention head selection followed by bias tuning. In the first step, we selected the top 160 attention heads based on their importance scores. In the second step, we refined our focus by selecting the top 32 heads specifically and tuning bias vectors at these locations. The $l1$-regularization coefficient was set to 0.005 in the first step. Note that LoFiT was only applied during the second stage of training, as it is designed for personalized adaptation.

**Representation-Based Methods (ReFT and `PerFit`)** For our representation-based methods, we conducted a random search across 20 trials to explore various hyperparameter combinations due to computational constraints. The search space, detailed in Table 8, included low-rank dimensions ranging from 4 to 64, user low-rank dimensions from 4 to 64, different intervention layer combinations (single layer "15" to multiple layers "14;15;16;17;18"), various position configurations (from "f7+l7" to "l5"), and maximum epochs ranging from 2 to 6. To ensure fair comparison between ReFT and our proposed `PerFit` method, we used identical hyperparameters with the exception that ReFT's rank dimension was set to the sum of our global and local ranks, maintaining equivalent parameter counts. All other hyperparameters for ReFT, including intervention layers, positions, and maximum epochs, were kept consistent with our optimal configurations. The optimal hyperparameter configurations determined through our experiments are presented in Table 9[8].

### D.6 Dataset Descriptions and Prompts

For detailed dataset composition and descriptions, please refer to LaMP[9] (Salemi et al., 2023). OPPU presents a specific setting on the LaMP dataset, and its prompts can be found in OPPU[10] (Tan et al., 2024b).

---

[8]Note that due to computational resource constraints, our hyperparameter search was limited to a restricted space with a maximum of 20 trials. While these configurations represent the best results found within our search space, they may not be globally optimal. There remains significant potential for further parameter reduction and performance improvement through more extensive hyperparameter exploration.

[9]LaMP benchmark official website with dataset download links: https://lamp-benchmark.github.io/download

[10]OPPU official GitHub repository containing implementation details and prompt templates: https://github.com/TamSiuhin/OPPU

Table 8: Hyperparameter search space for representation-based methods

| Hyperparameter | Values |
|---|---|
| Low-Rank Dimension | {4, 8, 16, 32, 64} |
| User Low-Rank Dimension | {4, 8, 16, 32, 64} |
| Intervention Layers (sep. w/ ';') | {"15", "14;15", "14;15;16", "14;15;16;17;18"} |
| Positions (Prefix+Suffix) | {"f7+l7", "f5+l5", "l7", "l5"} |
| Maximum Epochs | {2, 4, 6} |

Table 9: Optimal hyperparameter configurations for representation-based methods (*LRank*: Low-Rank dim, *ULRank*: User Low-Rank dim, *IL*: Intervention Layers, *PL*: Position Layers, *EP*: EPochs) across tasks.

| Task | LRank | ULRank | IL | PL | EP |
|---|---|---|---|---|---|
| News Categorization | 16 | 32 | 14;15;16 | f7+l7 | 6 |
| Movie Tagging | 32 | 4 | 14;15 | f7+l7 | 4 |
| Product Rating | 32 | 4 | 14;15;16 | l7 | 4 |
| News Headline Generation | 32 | 4 | 14;15;16 | l7 | 4 |
| Scholarly Title Generation | 32 | 8 | 15 | f7+l7 | 4 |
| Tweet Paraphrasing | 32 | 16 | 14;15 | f7+l7 | 4 |

# E  SUPPLEMENTARY EXPERIMENTAL RESULTS

In this section, we present comprehensive experimental results to complement our main findings, focusing on answering the following key research questions:

1. How stable and statistically significant are our performance improvements across multiple runs (Appendix E.1)?

2. What is the optimal layer for intervention (Appendix E.2)?

3. How does our method compare with collective LoRA approaches in terms of efficiency and effectiveness (Appendix E.3)?

4. How robust is our method across different backbone models (Appendix E.4)?

5. How well does our method perform in cold-start scenarios with limited collective users (Appendix E.5)?

6. How transferable is the collective shift across different users and tasks (Appendix E.6)?

7. How does our method affect the model's general deployment capability (Appendix E.7)?

## E.1  PERFORMANCE STABILITY ANALYSIS

Table 10 shows the mean and standard deviation of our method's performance across multiple runs on all tasks. Our method demonstrates statistically significant improvements ($p < 0.05$) over OPPU in several key metrics including LaMP-2N Accuracy, LaMP-2M Accuracy and F1, LaMP-4 R-1, and LaMP-5 R-L.

## E.2  MORE ANALYSIS ABOUT LAYER-WISE INTERVENTION

Table 11 presents a comprehensive comparison of model performance when interventions are applied at different transformer layers. We observe that intervening at lower to middle layers (e.g., layers 5 and 10) yields the best results across both classification and generation tasks. Specifically, performance peaks around layer 10 for Movie Tagging and News Categorization, and around layer 5 for News Headline Generation. As the intervention moves to higher layers, performance consistently declines, with a dramatic drop at the final layer 31, where the model fails to properly format and respond to the query according to the specified instructions. This trend suggests that user-specific information is most effectively injected and utilized in the earlier and middle layers of the model, while late-layer

Table 10: Comparison of our method and OPPU results (mean $\pm$ std) with statistical significance (p-value) across all tasks.

| | LaMP-2N | | LaMP-2M | | LaMP-3 | |
|---|---|---|---|---|---|---|
| | Acc $\uparrow$ | F1 $\uparrow$ | Acc $\uparrow$ | F1 $\uparrow$ | MAE $\downarrow$ | RMSE $\downarrow$ |
| Ours | $\mathbf{0.818}_{\pm 3.2e-5}$ | $0.586_{\pm 2.8e-5}$ | $\mathbf{0.630}_{\pm 2.5e-5}$ | $\mathbf{0.518}_{\pm 5.4e-5}$ | $\mathbf{0.179}_{\pm 5.2e-3}$ | $\mathbf{0.443}_{\pm 5.9e-3}$ |
| OPPU | $0.810_{\pm 3.2e-3}$ | $\mathbf{0.589}_{\pm 6.3e-3}$ | $0.600_{\pm 3.4e-3}$ | $0.493_{\pm 6.3e-3}$ | $\mathbf{0.179}_{\pm 5.2e-3}$ | $\mathbf{0.443}_{\pm 5.9e-3}$ |
| p-value | 0.026 | 0.089 | 0.007 | 0.003 | 0.256 | 0.632 |

| | LaMP-4 | | LaMP-5 | | LaMP-7 | |
|---|---|---|---|---|---|---|
| | R-1 $\uparrow$ | R-L $\uparrow$ | R-1 $\uparrow$ | R-L $\uparrow$ | R-1 $\uparrow$ | R-L $\uparrow$ |
| Ours | $\mathbf{0.207}_{\pm 9.1e-5}$ | $\mathbf{0.186}_{\pm 6.2e-5}$ | $\mathbf{0.521}_{\pm 3.3e-3}$ | $\mathbf{0.451}_{\pm 2.5e-3}$ | $0.525_{\pm 3.1e-3}$ | $0.472_{\pm 2.3e-3}$ |
| OPPU | $0.191_{\pm 5.3e-3}$ | $0.171_{\pm 9.6e-3}$ | $0.519_{\pm 1.2e-2}$ | $0.442_{\pm 6.7e-3}$ | $\mathbf{0.539}_{\pm 1.0e-2}$ | $\mathbf{0.483}_{\pm 5.2e-3}$ |
| p-value | 0.024 | 0.055 | 0.089 | 0.011 | 0.401 | 0.119 |

interventions may disrupt the learned representations or fail to propagate personalization signals. These findings highlight the importance of carefully selecting the intervention layer to maximize the benefits of representation-level personalization and align with our empirical layer-wise study about the low-rank pattern and collective shift in Appendix B.5.

Table 11: Layer-wise intervention results across three representative tasks: Movie Tagging (LaMP-2M), News Categorization (LaMP-2N), and News Headline Generation (LaMP-4). For classification tasks, we report Accuracy (Acc) and F1 score; for the generation task, we report ROUGE-1 (R-1) and ROUGE-L (R-L). Each row corresponds to interventions at a specific transformer layer. This table illustrates how the effectiveness of personalization varies with the intervention layer. Best results are highlighted in **bold**.

| Layer | Movie Tagging | | News Categorization | | News Headline Gen. | |
|---|---|---|---|---|---|---|
| | Acc $\uparrow$ | F1 $\uparrow$ | Acc $\uparrow$ | F1 $\uparrow$ | R-1 $\uparrow$ | R-L $\uparrow$ |
| 0 | 0.598 | 0.479 | 0.809 | 0.599 | 0.207 | 0.186 |
| 5 | **0.630** | 0.522 | 0.808 | 0.593 | **0.208** | **0.187** |
| 10 | 0.634 | **0.523** | **0.817** | **0.603** | 0.203 | 0.183 |
| 15 | 0.616 | 0.506 | 0.805 | 0.587 | 0.190 | 0.171 |
| 20 | 0.609 | 0.486 | 0.796 | 0.569 | 0.170 | 0.153 |
| 25 | 0.590 | 0.468 | 0.795 | 0.560 | 0.161 | 0.145 |
| 31 | 0.000 | 0.000 | 0.000 | 0.000 | 0.031 | 0.030 |

> Our layer-wise intervention analysis reveals a "sweet spot" in transformer architecture: personalization signals are most effective when injected in the middle layers (5-10), while higher-layer interventions can be detrimental. This finding not only guides optimal intervention placement but also suggests that personalization is fundamentally a mid-level representation learning problem, rather than a high-level semantic or low-level feature adaptation task.

### E.3 COLLECTIVE LoRA AND PERSONALIZED PERFIT

Table 12 presents a comprehensive comparison between our PerFit approach and two variants of collective LoRA with PerFit-P. The results reveal several interesting patterns: First, while LoRA-C (all) + PerFit-P achieves the best performance across most metrics, the improvement over our original PerFit method is relatively modest, particularly for News Categorization and News Headline Generation. This suggests that our layer-wise intervention approach already captures most of the benefits of personalization. Second, LoRA-C (partial) + PerFit-P generally performs worse than both other approaches, indicating that selective layer fine-tuning may not be as effective as either full fine-tuning or our targeted intervention approach. Finally, the parameter ratios show that our method achieves competitive performance while using significantly fewer parameters (8-16x fewer), highlighting its efficiency in terms of both computational resources and storage requirements.

Table 12: Comparison of different personalization approaches across four representative tasks. We evaluate our method (`PerFit`), collective LoRA with `PerFit`-P where the first-stage fine-tuning layers match `PerFit`'s intervention layers (LoRA-C (partial)), and collective LoRA with `PerFit`-P where all layers are fine-tuned (LoRA-C (all)). The *Param. Ratio* shows the collective parameter ratio between the `PerFit` and the LoRA-C (all). Best results are highlighted in **bold**.

| Method | LaMP-2N | | LaMP-2M | |
|---|---|---|---|---|
| | Acc ↑ | F1 ↑ | Acc ↑ | F1 ↑ |
| **PerFit** | **0.818** | **0.586** | 0.630 | 0.518 |
| LoRA-C (partial) + PerFit-P | 0.788 | 0.582 | 0.604 | 0.467 |
| LoRA-C (all) + PerFit-P | 0.791 | 0.584 | **0.640** | **0.529** |
| *Param. Ratio* | 1/16 | | 1/12 | |

| Method | LaMP-4 | | LaMP-7 | |
|---|---|---|---|---|
| | R-1 ↑ | R-L ↑ | R-1 ↑ | R-L ↑ |
| **PerFit** | **0.207** | **0.186** | 0.525 | 0.472 |
| LoRA-C (partial) + PerFit-P | 0.193 | 0.173 | 0.509 | 0.461 |
| LoRA-C (all) + PerFit-P | 0.205 | **0.186** | **0.561** | **0.515** |
| *Param. Ratio* | 1/8 | | 1/12 | |

> This finding challenges the conventional wisdom that more parameters necessarily lead to better personalization, suggesting that strategic intervention at key layers may be the key to efficient and effective model adaptation.

### E.4 BACKBONE ABLATION STUDY

To give a more comprehensive understanding of our method's effectiveness, we conducted an ablation study using different backbone models, including LLaMA3-8B, Qwen2.5-7B, LLaMA3-3B, and LLaMA3-1B. We compared our method (`PerFit`) against other personalization approaches such as LoRA-C, LoRA-P, and OPPU across four representative tasks: Movie Tagging, News Categorization, News Headline Generation, and Tweet Paraphrasing. For simplicity, we maintained consistent hyperparameter configurations across all backbone models in our `PerFit` experiments: we set both the low-rank dimension and user low-rank dimension to 32, applied interventions at layers 7, 9, and 11, used both front and last 7 positions (f7+l7) for intervention, and trained for 2 epochs. We choose these settings considering the balance between performance and computational efficiency, which can be further optimized for each specific backbone. The uniformity in hyperparameters ensures that performance differences are attributable to the backbone model rather than tuning variations. This is also a practical choice for followers who may not have the resources for extensive hyperparameter tuning for each model.

The results, summarized in Table 13. Notably, the performance gap between `PerFit` and other methods widens as the model size decreases, indicating that our approach is particularly effective for smaller models where parameter efficiency is crucial. This suggests that `PerFit` not only enhances personalization but also maintains robustness across varying model capacities.

### E.5 COLD-START WITH LIMITED COLLECTIVE USERS

To evaluate our method's robustness in cold-start scenarios and data-scarce conditions, we conducted experiments with limited collective users in the first training stage. This setting simulates real-world scenarios where new users arrive and only limited collective data is available for learning the initial representation shift. We compared performance with varying numbers of collective users (10 and 100) against OPPU across multiple tasks.

Table 13: Performance comparison across different backbone models and personalization methods. Trainable parameters and percentages are also reported.

| Method | LaMP-2M | | LaMP-2N | | LaMP-4 | | LaMP-7 | | #Param | %Param |
|---|---|---|---|---|---|---|---|---|---|---|
| | Acc | F1 | Acc | F1 | R-1 | R-L | R-1 | R-L | | |
| **LLaMA3-8B** | | | | | | | | | | |
| LoRA-C | 0.537 | 0.494 | 0.798 | 0.554 | 0.201 | 0.181 | 0.553 | 0.501 | 4.72M | 0.059% |
| LoRA-P | 0.212 | 0.125 | 0.539 | 0.333 | 0.094 | 0.086 | 0.136 | 0.120 | 3.41M | 0.042% |
| OPPU | 0.627 | 0.529 | 0.802 | 0.587 | 0.209 | 0.188 | 0.494 | 0.442 | 8.13M | 0.051% |
| PerFit | 0.644 | 0.550 | 0.818 | 0.612 | 0.201 | 0.180 | 0.533 | 0.487 | 1.57M | 0.010% |
| **Qwen2.5-7B** | | | | | | | | | | |
| LoRA-C | 0.429 | 0.384 | 0.792 | 0.519 | 0.177 | 0.159 | 0.526 | 0.469 | 3.44M | 0.045% |
| LoRA-P | 0.485 | 0.305 | 0.781 | 0.560 | 0.164 | 0.147 | 0.446 | 0.400 | 2.52M | 0.033% |
| OPPU | 0.567 | 0.451 | 0.789 | 0.555 | 0.185 | 0.166 | 0.524 | 0.473 | 5.96M | 0.039% |
| PerFit | 0.612 | 0.508 | 0.808 | 0.612 | 0.182 | 0.164 | 0.532 | 0.482 | 1.38M | 0.009% |
| **LLaMA3-3B** | | | | | | | | | | |
| LoRA-C | 0.438 | 0.401 | 0.783 | 0.510 | 0.175 | 0.156 | 0.534 | 0.477 | 3.21M | 0.100% |
| LoRA-P | 0.163 | 0.063 | 0.688 | 0.501 | 0.124 | 0.110 | 0.201 | 0.171 | 2.29M | 0.071% |
| OPPU | 0.579 | 0.464 | 0.792 | 0.566 | 0.185 | 0.165 | 0.512 | 0.460 | 5.51M | 0.086% |
| PerFit | 0.644 | 0.550 | 0.818 | 0.612 | 0.201 | 0.180 | 0.533 | 0.487 | 1.18M | 0.018% |
| **LLaMA3-1B** | | | | | | | | | | |
| LoRA-C | 0.402 | 0.349 | 0.771 | 0.491 | 0.147 | 0.132 | 0.524 | 0.474 | 1.18M | 0.095% |
| LoRA-P | 0.105 | 0.041 | 0.490 | 0.309 | 0.075 | 0.067 | 0.160 | 0.143 | 0.85M | 0.069% |
| OPPU | 0.529 | 0.405 | 0.780 | 0.545 | 0.157 | 0.141 | 0.376 | 0.338 | 2.03M | 0.082% |
| PerFit | 0.563 | 0.448 | 0.796 | 0.577 | 0.146 | 0.130 | 0.500 | 0.456 | 0.79M | 0.032% |

Table 14: Performance comparison under limited collective users. Results show both OPPU and our method (PerFit) with 10 and 100 users in the first training stage.

| Method | LaMP-2N | | LaMP-2M | | LaMP-4 | | LaMP-7 | |
|---|---|---|---|---|---|---|---|---|
| | Acc | F1 | Acc | F1 | R-1 | R-L | R-1 | R-L |
| 10 (OPPU) | 0.783 | 0.569 | 0.168 | 0.075 | 0.197 | 0.177 | 0.142 | 0.136 |
| 10 (PerFit) | 0.800 | 0.589 | 0.535 | 0.398 | 0.199 | 0.179 | 0.465 | 0.418 |
| 100 (OPPU) | 0.801 | 0.600 | 0.538 | 0.385 | 0.197 | 0.177 | 0.472 | 0.428 |
| 100 (PerFit) | 0.812 | 0.598 | 0.587 | 0.464 | 0.201 | 0.181 | 0.476 | 0.429 |

As shown in Table 14, our method demonstrates remarkable resilience under data scarcity. Even with only 10 collective users, PerFit maintains stable performance across all tasks, while OPPU experiences significant degradation, particularly in Movie Tagging (LaMP-2M) and Tweet Paraphrasing (LaMP-7). This robustness can be attributed to our approach's focus on learning approximate common biases in the representation space during the first stage, rather than attempting to capture fine-grained user patterns.

The results suggest that our method is particularly well-suited for real-world applications where collecting extensive collective user data may be challenging or impractical. The stable performance under limited data conditions also indicates that the learned representation shifts effectively capture fundamental patterns that generalize well to new users.

Beyond the number of collective users, we further investigate how sensitive PerFit is to the specific composition and behavioral diversity of the Stage-1 user set. To this end, we randomly sample five different user groups for Stage-1 (each containing the same number of users), learn a collective shift for each group, and then fine-tune on a fixed set of unseen users in Stage-2. Table 15 reports the downstream performance for Movie Tagging, News Categorize, and Tweet Paraphrase across these user batches.

Table 15: Sensitivity of `PerFit` to Stage-1 collective user composition. We report results on a fixed set of unseen users when the Stage-1 collective shift is learned on five different randomly sampled user batches (each with the same number of users). The variance across batches is extremely small, indicating that `PerFit` is robust to the specific choice of collective users.

| User batch | Movie Tagging | | News Categorization | | Tweet Paraphrasing | |
|---|---|---|---|---|---|---|
| | Acc | F1 | Acc | F1 | R-1 | R-L |
| 0 | 0.600 | 0.478 | 0.804 | 0.588 | 0.482 | 0.440 |
| 1 | 0.604 | 0.485 | 0.808 | 0.600 | 0.489 | 0.445 |
| 2 | 0.601 | 0.482 | 0.804 | 0.591 | 0.488 | 0.446 |
| 3 | 0.599 | 0.478 | 0.809 | 0.595 | 0.482 | 0.442 |
| 4 | 0.597 | 0.476 | 0.810 | 0.592 | 0.475 | 0.433 |
| Variance | $5.0 \times 10^{-6}$ | $1.1 \times 10^{-5}$ | $6.0 \times 10^{-6}$ | $1.7 \times 10^{-5}$ | $2.5 \times 10^{-5}$ | $2.1 \times 10^{-5}$ |

These results show that the performance of `PerFit` remains remarkably stable across different Stage-1 user batches, with only negligible variance in both classification and generation metrics. This suggests that the learned collective shift captures a dominant, population-level direction in the representation space, which is preserved even when the underlying user subset is resampled. Consequently, Stage-2 personalization operates on top of a robust shared structure, and `PerFit` is not overly sensitive to the particular composition or behavioral distribution of the collective user set, consistent with our analysis of the representation geometry in Section 3 and Appendix B.6.

## E.6 Transferability of the Collective Shift Across Users and Tasks

The "collective shift" in our framework captures a robust, population-level personalization trend that generalizes across both users and tasks, rather than merely acting as a domain-specific adaptation. This is primarily evidenced by its transferability: the Stage-1 collective shift directly benefits previously unseen users in Stage-2 without retraining, and further proves effective even when transferred across different tasks. As shown in Table 16, a collective shift learned on one task remains functional for distinct downstream tasks, suggesting it encodes fundamental personalization characteristics beyond single-dataset statistics.

Crucially, this transfer behavior is consistent with the geometric structure of the learned representations analyzed in Appendix B.6. We observe that tasks located closer in the $\delta$-space (e.g., Movie Tagging and News Categorization) exhibit smoother transfer, whereas pairs with large geometric separation (e.g., Tweet Paraphrase and News Headline) show a more pronounced performance drop. This alignment confirms that the collective shift captures structural personalization patterns that are often shared across related domains.

Table 16: Cross-task transferability of the collective shift. The rows indicate the task used for Stage-1 training, while the columns indicate the task used for Stage-2 training and evaluation.

| Stage-1 Task | Movie Tagging | | News Categorization | | News Headline | | Tweet Paraphrasing | |
|---|---|---|---|---|---|---|---|---|
| | Acc | F1 | Acc | F1 | R-1 | R-L | R-1 | R-L |
| Movie Tagging | 0.636 | 0.520 | 0.815 | 0.591 | 0.194 | 0.174 | 0.466 | 0.420 |
| News Categorization | 0.545 | 0.408 | 0.818 | 0.588 | 0.195 | 0.175 | 0.467 | 0.419 |
| News Headline | 0.540 | 0.381 | 0.814 | 0.586 | 0.207 | 0.186 | 0.212 | 0.203 |
| Tweet Paraphrasing | 0.521 | 0.364 | 0.795 | 0.579 | 0.196 | 0.176 | 0.490 | 0.447 |

These results show that the Stage-1 collective shift exhibits non-trivial cross-task transferability, suggesting a connection to task-level personalization characteristics rather than a single dataset. This observation is consistent with our $\delta$-vectors analysis and points to an interesting direction for future work on understanding when and how such collective shifts transfer across tasks by exploring benchmarks with more complex and diverse tasks.

Table 17: Impact of `PerFit` personalization on commonsense reasoning benchmarks (Accuracy ↑). `PerFit` is applied to user-level datasets (Movie Tagging and Tweet Paraphrase), and the resulting personalized models are evaluated on standard commonsense tasks. Performance differences between Base and PerFit are negligible.

| Model | BoolQ | PIQA | SIQA | HellaS | WinoG | ARC-e | ARC-c | OBQA | Avg. |
|---|---|---|---|---|---|---|---|---|---|
| **Qwen2.5-7B** (Movie Tagging) | | | | | | | | | |
| Base | 0.846 | 0.787 | 0.547 | 0.600 | 0.730 | 0.804 | 0.479 | 0.336 | 0.641 |
| PerFit | 0.846 | 0.787 | 0.548 | 0.600 | 0.732 | 0.804 | 0.482 | 0.338 | 0.642 |
| **Llama-2-7B** (Tweet Paraphrasing) | | | | | | | | | |
| Base | 0.777 | 0.781 | 0.461 | 0.572 | 0.690 | 0.763 | 0.434 | 0.314 | 0.599 |
| PerFit | 0.764 | 0.784 | 0.459 | 0.571 | 0.691 | 0.759 | 0.424 | 0.316 | 0.596 |

### E.7 IMPACT OF PERSONALIZATION INTERVENTIONS ON GENERAL CAPABILITIES

`PerFit` injects low-rank personalization interventions at multiple transformer layers, including relatively early layers, which raises a natural concern about potential interference with the model's generic language understanding and reasoning abilities. To empirically assess this, we evaluate frozen personalized models (after Stage-2) on widely used general-purpose benchmarks, including commonsense reasoning tasks (BoolQ, PIQA, SIQA, HellaSwag, WinoGrande, ARC-e, ARC-c, OBQA) . The personalized models are obtained by applying `PerFit` to different user-facing datasets (Movie Tagging for Qwen2.5-7B and Tweet Paraphrase for Llama-2-7B), while the underlying backbone weights remain frozen after Stage-2.[11]

Formally, we compare the following two settings: (i) Base, the original pretrained model without any personalization, and (ii) `PerFit`, the same backbone with the learned low-rank interventions activated after Stage-2 personalization. As summarized in Table 17, the performance differences between Base and `PerFit` are consistently negligible across all benchmarks, with fluctuations well within typical evaluation noise. These results indicate that `PerFit`'s low-rank interventions effectively preserve the backbone's generic capabilities in commonsense reasoning and natural language understanding, while still enabling strong user-level personalization on downstream tasks.

Overall, these evaluations provide empirical evidence that multi-layer personalization via PerFit does not come at the expense of the model's generic capabilities on standard benchmarks. The learned interventions remain sufficiently localized in representation space, enabling user-specific adaptation while preserving the backbone's broader linguistic and reasoning competence.

## F BROADER IMPACTS

Personalized Large Language Models (LLMs), particularly through methods like fine-tuning in representation space, offer transformative potential for human-computer interaction and information access. This approach, by subtly adapting LLMs via their underlying representation space rather than full model retraining, significantly enhances resource efficiency and scalability, making deep personalization feasible for a broader range of applications and users. This personalized tailoring promises to revolutionize user experience by matching communication style, vocabulary, and level of detail to individual needs, improving efficiency in tasks, and fostering hyper-personalized learning. Such adaptation inherently boosts accessibility, bridging communication gaps for diverse users.

However, these advancements come with substantial ethical and societal challenges. Personalization, even when achieved in representation space, risks implicitly encoding sensitive user information, raising significant privacy and data security concerns through potential re-identification. It can also inadvertently amplify existing biases within training data, leading to skewed or discriminatory information delivery. Furthermore, by constantly curating content to individual preferences, personalized LLMs can create restrictive "filter bubbles" and "echo chambers," limiting exposure to diverse viewpoints and potentially enabling sophisticated, personalized misinformation or manipulation

---

[11]We follow the standard evaluation protocol and prompts of each benchmark.

campaigns that could subtly erode user autonomy. The opaque nature of these models, further compounded by personalization, also complicates transparency and explainability.

Addressing these critical impacts requires a proactive, multi-faceted approach. Robust privacy-preserving techniques, comprehensive bias detection and mitigation strategies, and transparent user controls are essential. Developing clear ethical guidelines and fostering interdisciplinary collaboration among researchers, ethicists, and policymakers are crucial steps to ensure that PLLMs are developed and deployed responsibly, maximizing their societal benefits while minimizing potential harms.

## G   USE OF LLMS

This work utilizes large language models (LLMs), including ChatGPT `https://chat.openai.com/` and Claude Code `https://claude.ai/chats`, as general-purpose tools for writing assistance. These tools were employed for polishing text, correcting formatting errors, and checking grammar throughout the writing process.

This use of LLMs complies with the ICLR 2026 Author Guide `https://iclr.cc/Conferences/2026/AuthorGuide`, ensuring adherence to the conference's guidelines for AI assistance in academic writing.

