# OpenReview forum: "PerFit: Exploring Personalization Shifts in Representation Space of LLMs"
_ICLR.cc/2026/Conference — ICLR 2026 Poster_

### Official Review · Reviewer_eWQo · 2025-10-29

**Soundness:** 3
**Presentation:** 4
**Contribution:** 3
**Rating:** 6
**Confidence:** 3

**Summary:**

This paper presents PerFit, a novel two-stage personalized fine-tuning method based on activation engineering that directly intervenes in the hidden representation space of large language models. The method design is motivated by an analytical study on difference-in-means personalization vectors, revealing that personalization lies in a low-rank subspace with collective and user-specific shifts. On top of this, PerFit first learns a global “collective shift” shared by all users and then learn user-specific vectors to modify the hidden representations. Experiments on various benchmarks show that PerFit achieves superior performance compared to state-of-the-art PEFT baselines, while reducing personalized parameters by ~92%, demonstrating the method’s efficiency and scalability.

**Strengths:**

1. The paper is well-presented and easy for readers to understand.

2. The analytical investigation on personalization vectors is interesting and well motivates the two-stage design of the method.

3. By operating on low-rank hidden activation, PerFit greatly reduces the cost of both training and storage for each user, opening a new door for the following social simulation research, which involves many user behavior simulations.

4. The experiments are comprehensive and well demonstrate the competitive performance and efficiency across six personalization tasks.

**Weaknesses:**

1. While the experiments are comprehensive, the impact of collective user composition in Stage-1 remains underexplored. Since Stage-1 requires training across a group of users, it is likely that different levels of user diversity or distributional imbalance would affect the effectiveness and stability of the learned collective shift. The paper does not analyze the sensitivity of PerFit to the selection and variety of collective users.

2. Although PerFit generally performs well, the paper provides limited insight into cases where it underperforms. For example, in Table 3, PerFit is worse than OPPU on Tweet Paraphrasing, yet no analysis is provided. Understanding failure patterns would be valuable for both practical deployment and method improvement.

3. Efficiency claims mainly rely on parameter size and relative training time relationships. However, key metrics for real-world deployment, such as wall-clock latency (training and inference), GPU memory consumption, are not reported. Thus, the practical scalability advantage remains partially demonstrated.

4. The two-stage design assumes that the learned collective shift generalizes across users. However, in dynamic environments where new users continuously join, it is unclear whether Stage-1 must be retrained or can be directly transferred. The paper does not empirically validate the stability of Stage-1 when user distributions change.

5. PerFit injects personalization interventions into multiple layers, including early layers that encode core linguistic and semantic representations. Without any mechanism to constrain interference, such interventions may degrade the model’s general abilities. Yet no evaluation is conducted to verify preservation of generic performance or robustness.


The paper is technically strong and practically impactful, with clear novelty in modeling personalization in the representation space. Some analyses require further expansion, but the work provides valuable contributions to personalized LLM adaptation and opens promising directions for future research.

**Questions:**

1. Can the authors provide insights or evaluate how the performance varies when the collective user set differs in diversity or behavioral distribution?

2. What are the author’s hypotheses on why PerFit underperforms in the Tweet Paraphrasing task, and are there task-specific characteristics where representation-space interventions may be less effective?

3. Could the authors provide more detailed runtime and memory measurements to substantiate the real-world efficiency improvements?

4. Given dynamic real-world personalization, the transferability of Stage-1 to unseen users could be an interesting direction to explore. Have the authors observed whether the collective shift generalizes across user populations?

5. Since personalization is applied at several (including early) layers, evaluating the effect on general model capability might provide additional assurance. Have the authors examined whether general capabilities are impacted when personalization vectors are injected?

---

> ### Author Response · Authors · 2025-11-21
> **Rebuttal (1/3)**
>
> We sincerely thank the reviewer for the positive and encouraging feedback and appreciate the recognition of *our paper’s presentation quality, analytical investigation, methodological motivation, and practical impact*. Below, **W** denotes weaknesses, **Q** denotes questions, and **R** provides our responses.
>
> &nbsp;
>
> ---
>
> **W1 & Q1:** Sensitivity of PerFit to the collective user set.
>
> **R:** Thank you for raising this important question. **We conducted an ablation study to directly examine how sensitive PerFit is to the composition and diversity of the Stage-1 collective user set.** In this experiment, we *randomly sampled 5 different user groups* for Stage-1 (each containing the same number of users), trained a collective shift for each group, and then fine-tuned this model separately on a fixed set of unseen users. The results are shown below:
>
> | user_batch_index | Movie Tagging_acc | Movie Tagging_F1 | News Categorize_acc | News Categorize_F1 | Tweet Paraphrase_R-1 | Tweet Paraphrase_R-L |
> | --- | --- | --- | --- | --- | --- | --- |
> | 0 | 0.600 | 0.478 | 0.804 | 0.588 | 0.482 | 0.440 |
> | 1 | 0.604 | 0.485 | 0.808 | 0.600 | 0.489 | 0.445 |
> | 2 | 0.601 | 0.482 | 0.804 | 0.591 | 0.488 | 0.446 |
> | 3 | 0.599 | 0.478 | 0.809 | 0.595 | 0.482 | 0.442 |
> | 4 | 0.597 | 0.476 | 0.810 | 0.592 | 0.475 | 0.433 |
> | **Variance** | **5.0 × 10⁻⁶** | **1.1 × 10⁻⁵** | **6.0 × 10⁻⁶** | **1.7 × 10⁻⁵** | **2.5 × 10⁻⁵** | **2.1 × 10⁻⁵** |
>
> This stability indicates that PerFit **is not highly sensitive to the specific composition or behavioral distribution of the Stage-1 user set**. The reason is that the learned collective shift corresponds to a **dominant population-level direction** in representation space (as shown in Section 3 and Appendix B.6), which remains stable even under resampling of user subsets. In other words, sampling different user groups perturbs the collective shift only slightly, and Stage-2 adapts effectively on top of this stable shared structure.
>
> &nbsp;
>
> ---
>
> **W2 & Q2:** Understanding PerFit’s underperformance on Tweet Paraphrasing.
>
> **R:** Thanks very much for the valuable suggestion of analyzing cases where PerFit underperforms. *Tweet Paraphrasing* is indeed a special case and differs structurally from the other LaMP tasks.
>
> The supervision signal for personalization is fundamentally different. In tasks such as *Scholarly Title* or *News Headline*, each historical example provides a **paired input–output transformation** (e.g., `abstract → title`, `article → headline`). The corresponding δ-vectors therefore capture how a user consistently transforms source content into personalized outputs. In *Tweet Paraphrasing*, however, users’ histories contain **only their past tweets**, without the original source tweets they paraphrased. Yet the downstream task requires rewriting *new* input tweets. As a result, PerFit must estimate its representation shift **from “standalone style samples” rather than paired input–output transformations**, which tend to encode topic preferences or surface-level noise rather than a coherent paraphrasing behavior.
>
> Our analysis in *Appendix B.6* further supports this distinction: the tweet dataset exhibits a **unique and more dispersed δ-vector geometry**, with personalization directions that deviate substantially from those of other tasks. This reflects the weaker and less structured personalization signal available for this dataset. In such a regime, the available signal provides less structure for representation-level updating, whereas OPPU can accommodate this variability through its larger parameter budget.
>
> These observations explain why PerFit performs slightly below OPPU on this specific task and suggest that **tasks lacking paired input–output personalization signals may benefit from additional contrastive signals or hybrid designs.** Importantly, PerFit remains competitive and outperforms the other LoRA-based baselines, demonstrating robustness even under dispersed personalization signals. We will integrate this analysis into the revised version to improve clarity.
>
> ---

---

> ### Author Response · Authors · 2025-11-21
> **Rebuttal (2/3)**
>
> **W3 & Q3:**  Runtime and memory measurements for real-world efficiency.
>
> **R:** Thank you for the valuable suggestion. We agree that real-world deployability requires examining **wall-clock latency** and **GPU memory usage** in addition to parameter counts. We have therefore included detailed runtime and memory measurements under identical hardware settings (A6000-48GB).
>
> Across all tasks, PerFit consistently demonstrates **lower GPU memory usage**, **lower or comparable total training latency**, and **substantially faster inference** relative to the strong LoRA-based OPPU baseline. These results corroborate the practical scalability advantages suggested by our parameter-efficiency analysis.
>
> - **GPU Memory Usage (GB)**
>
> | Dataset | Movie Tagging | Movie Tagging | News Categorize | News Categorize | News Headline | News Headline | Product Rating | Product Rating | Scholarly Title | Scholarly Title | Tweet Paraphrase | Tweet Paraphrase |
> | --- | --- | --- | --- | --- | --- | --- | --- | --- | --- | --- | --- | --- |
> | Metric | Training | Inference | Training | Inference | Training | Inference | Training | Inference | Training | Inference | Training | Inference |
> | OPPU | 31.705GB | 30.49GB | 28.765GB | 26.97GB | 27.48GB | 26.94GB | 29.355GB | 29.21GB | 31.745GB | 30.01GB | 28.545GB | 27.05GB |
> | PerFit | 25.865GB | 25.46GB | 23.175GB | 19.91GB | 20.165GB | 18.97GB | 18.91GB | 21.51GB | 19.65GB | 21.00G | 19.34GB | 17.47GB |
>
> - **Wall-Clock Latency (training & inference)**
>
> | Dataset | Movie Tagging | Movie Tagging | News Categorize | News Categorize | News Headline | News Headline | Product Rating | Product Rating | Scholarly Title | Scholarly Title | Tweet Paraphrase | Tweet Paraphrase |
> | --- | --- | --- | --- | --- | --- | --- | --- | --- | --- | --- | --- | --- |
> | Metric | Training | Inference | Training | Inference | Training | Inference | Training | Inference | Training | Inference | Training | Inference |
> | OPPU | 20.59m | 8.66s | 22.86m | 37.84s | 40.19m | 30.84m | 10.51h | 17.78s | 5.87h | 8.28s | 1.50h | 17.45s |
> | PerFit | 12.51m | 4.44s | 15.06m | 22.17s | 21.72m | 1.62m | 5.89h | 0.24s | 2.14h | 0.75s | 41.26m | 1.15s |
>
> The results confirm that PerFit's two-stage design **not only reduces parameter storage but also yields meaningful improvements in memory footprint and runtime, strengthening its practical scalability.** We will incorporate these measurements into the revised version.
>
> &nbsp;
>
> ---
>
> **W4 & Q4:** Transferability of the Stage-1 collective shift to unseen user populations.
>
> **R:** Thank you for raising this important point. We clarify that PerFit is **explicitly designed for dynamic environments where new users continuously join, and our experimental setup already reflects this scenario.**
>
> Across all experiments, **Stage-2 personalization is always performed on users who never appear in Stage-1**. The two stages use **strictly disjoint user sets**, and the model never receives any joint training or user overlap between the stages. This corresponds to a cold-start personalization setting: Stage-1 learns a population-level trend once, and Stage-2 adapts it to entirely new users with small personal data.
>
> The consistently strong performance across six downstream tasks shows that the learned collective shift **transfers reliably to unseen user populations** without requiring any retraining of Stage-1. **This directly demonstrates that Stage-1 does not depend on the specific users it was trained on, and can be reused as new users arrive**—precisely the behavior desired in real-world personalization systems.
>
> ---

---

> > ### Author Response · Authors · 2025-11-21
> > **Rebuttal (3/3)**
> >
> > **W5 & Q5:** Impact of personalization interventions on general capabilities.
> >
> > **R:** Thanks very much for this excellent suggestion. We conducted additional evaluations to clarify the impact of personalization interventions on general capabilities. Specifically, we evaluated **frozen personalized models** (after Stage-2) on widely used general-purpose benchmarks, including **commonsense reasoning tasks** (BoolQ, PIQA, SIQA, HellaSwag, WinoGrande, ARC-e, ARC-c, OBQA) and **GLUE**.
> >
> > The evaluation protocol is straightforward:
> >
> > - **Base** = performance of the original pretrained model
> > - **PerFit** = performance after Stage-2 personalization trained on the same pretrained model using the dataset (e.g., Movie Tagging, Tweet Paraphrasing)
> >
> > Across all benchmarks, PerFit shows **negligible differences** compared to the base model, with fluctuations well within natural evaluation noise.  This indicates that the learned low-rank interventions preserve the backbone’s general reasoning and language understanding capabilities. Below we summarize the results:
> >
> > &nbsp;
> >
> > **Commonsense Datasets (Accuracy ↑)**
> >
> > * Qwen2.5-7B (Movie Tagging)
> >
> > | Model | BoolQ | PIQA | SIQA | HellaS | WinoG | ARC-e | ARC-c | OBQA | **Avg.** |
> > | --- | --- | --- | --- | --- | --- | --- | --- | --- | --- |
> > | **Base** | 0.846 | 0.787 | 0.547 | 0.600 | 0.730 | 0.804 | 0.479 | 0.336 | **0.641** |
> > | **PerFit** | 0.846 | 0.787 | 0.548 | 0.600 | 0.732 | 0.804 | 0.482 | 0.338 | **0.642** |
> >
> > - Llama-2-7B (Tweet Paraphrase)
> >
> > | Model | BoolQ | PIQA | SIQA | HellaS | WinoG | ARC-e | ARC-c | OBQA | **Avg.** |
> > | --- | --- | --- | --- | --- | --- | --- | --- | --- | --- |
> > | **Base** | 0.777 | 0.781 | 0.461 | 0.572 | 0.690 | 0.763 | 0.434 | 0.314 | **0.599** |
> > | **PerFit** | 0.764 | 0.784 | 0.459 | 0.571 | 0.691 | 0.759 | 0.424 | 0.316 | **0.596** |
> >
> > &nbsp;
> >
> > **GLUE (Accuracy ↑, CoLA = MCC, STSB = Pearson)**
> >
> > - Qwen2.5-7B (Movie Tagging)
> >
> > | Model | SST-2 | CoLA | MRPC | QQP | QNLI | MNLI | RTE | **Avg.** |
> > | --- | --- | --- | --- | --- | --- | --- | --- | --- |
> > | **Base** | 0.919 | 0.257 | 0.672 | 0.860 | 0.648 | 0.621 | 0.812 | **0.684** |
> > | **PerFit** | 0.919 | 0.256 | 0.672 | 0.860 | 0.648 | 0.620 | 0.809 | **0.683** |

---

### Official Review · Reviewer_xGBE · 2025-11-01

**Soundness:** 3
**Presentation:** 3
**Contribution:** 3
**Rating:** 6
**Confidence:** 3

**Summary:**

This paper introduces PerFit, a two-stage framework for personalizing large language models (LLMs) via interventions in the representation space rather than the parameter space. The authors first uncover that user-specific information lies in a low-rank subspace characterized by both a collective shift shared across users and personalized shifts unique to each user. Leveraging these findings, they propose a two-stage fine-tuning procedure that first learns a collective representation shift from all users, and then conducts fine-tuning personalized shifts for each user. Experimental results on six LaMP datasets show that PerFit achieves comparable or better performance to strong baselines, while significantly reducing the number of trainable parameters.

**Strengths:**

S1. Conducts a comprehensive investigation into how personalization is captured by LLMs towards tackling the challenges associated with balancing the efficiency and performance, which provides key findings, including that users share clear collective shifts leading to their two-stage framework.

S2. The framework itself is straightforward after uncovering the key findings in S1; although not technically complex, this is actually seen as a benefit.

S3. The practicality of the framework with being so parameter efficient is likely to have significant impact on the community and lead to numerous follow up works in this direction.

**Weaknesses:**

W1. Some concerns that with this framework primarily appearing to be heavily built on ReFT there is still something left for more novelty in the methodology.

W2. The collective and personalization shift hypothesis is supported primarily through visualizations rather than more rigorous analysis.

**Questions:**

Q1. Can you clarify the distinct differences between PerFit and prior representation fine-tuning methods?

Q2. How do training and inference times compare to those baseline methods, given that your projection operation could not be absorbed into the base model?

Q3. Do the collective shifts transfer across users or tasks, or are they tightly coupled to the dataset used in Stage 1?

---

> ### Author Response · Authors · 2025-11-21
> **Rebuttal (1/3)**
>
> We sincerely thank the reviewer for the positive and encouraging feedback and appreciate the recognition of our analysis of personalization in LLMs, the clarity of the proposed two-stage framework, and its parameter-efficient practical impact. Below, **W** denotes weaknesses, **Q** denotes questions, and **R** provides our responses.
>
> &nbsp;
>
> ---
>
> **W1 & Q1:**  Novelty relative to ReFT.
>
> **R:** Thank you for raising this point. While Stage-2 of our framework uses a ReFT-style intervention, **PerFit is not a variant of ReFT**, but a representation-centric personalization framework that is *motivated, structured, and justified* by our empirical discoveries about how personalization manifests in LLM representations.
>
> Our core contribution lies in **uncovering two key representational phenomena that had not been previously studied in the personalization literature**:
>
> (1) personalized information is captured in a **low-rank subspace**, and
>
> (2) personalization decomposes into a **collective shift** (shared across users) and a **personalized shift** (individual-specific)
>
> These findings motivate a **two-stage representation fine-tuning paradigm**: Stage-1 first learns the collective shift shared across the population, and Stage-2 then learns user-specific refinements *starting from the Stage-1 endpoint*, yielding a sequential and interpretable decomposition of personalization signals.
>
> This design is fundamentally different from ReFT. ReFT performs a single intervention directly on the pretrained model, whereas PerFit **explicitly leverages the collective-to-individual structure we discovered**. Without Stage-1, ReFT methods have no mechanism to capture the population-wide trend, and thus cannot realize the personalized decomposition that PerFit provides. Our experiments further show that this decomposition is not only conceptually meaningful but also crucial for parameter efficiency and performance.
>
> In summary, **the novelty of PerFit lies not in introducing a new intervention primitive but in establishing a principled, representation-level understanding of personalization and operationalizing it through a two-stage framework**. To the best of our knowledge, this is the first work that systematically analyzes personalization patterns in representation space and demonstrates that representation fine-tuning provides a natural and highly parameter-efficient solution for LLM personalization.
>
> We will further clarify this distinction and highlight the motivation and implications of our two-stage design in the revised version.
>
> ---

---

> > ### Author Response · Authors · 2025-11-21
> > **Rebuttal (2/3)**
> >
> > **W2:**  Support of collective and personalized shifts.
> >
> > **R:** Thank you for raising this point. We would first like to clarify that our analysis follows a **widely adopted and well-established analytical paradigm in mechanistic interpretability**, including activation steering and low-rank subspace identification [1, 2, 3]. These methods diagnose model behavior by examining consistent linear directions in activation space, and have become standard for identifying semantic, causal, and functional subspaces in LLMs. Our δ-vector framework is *directly aligned with this rigorous methodology rather than relying purely on visualization*.
> >
> > Building on this paradigm, our paper conducts an extensive and multi-perspective analysis of personalization structure. Section 3 quantitatively establishes the **low-rank nature** of δ-vectors through SVD, reports explained-variance ratios, and measures mean–variance patterns along principal components, showing a dominant shared direction with small dispersion. **Appendix B** further strengthens this with systematic robustness checks, such as repeating the analysis across layers and models, shuffled-controls showing the structure disappears without personalization, case-level inspection that aligns δ-vector positions with meaningful personalization behavior, and cross-task geometry revealing clear task-dependent organization rather than random variation. Collectively, these analyses provide strong and convergent evidence that the observed collective and personalized shifts are **robust, reproducible, and not artifacts of visualization**.
> >
> > In addition, we have added **Appendix B.5** (Page 21, highlighted in blue in the updated PDF), which provides a **layer-wise activation analysis.** This study shows that earlier and middle layers exhibit the most pronounced collective shift and lowest variance—consistent with prior findings that functional steering directions and semantic features tend to localize in these layers. Importantly, this pattern also aligns closely with our empirical performance observations in Section 4, offering a principled connection between representational structure and intervention effectiveness. This serves as further quantitative validation beyond visual inspection.
> >
> > We will highlight the quantitative evidence more prominently and include short statistical summaries alongside the figures in the next version.
> >
> > &nbsp;
> >
> > **References:**
> >
> > [1] Elhage, N., Nanda, N., Olah, C., et al. (2021). A Mathematical Framework for Transformer Circuits. Transformer Circuits Thread, Anthropic.
> >
> > [2] Meng, K., Bau, D., Andonian, A., & Liu, S. (2022). Locating and Editing Factual Associations in GPT. Advances in Neural Information Processing Systems (NeurIPS 2022).
> >
> > [3] Turner, A., Schaeffer, R., et al. (2023). Activation Addition: Steering Language Models Without Optimization. Alignment Research Center Technical Report.
> >
> > &nbsp;
> >
> > ---
> >
> > **Q2:** Training and inference time.
> >
> > **R:** Thank you for the question. We report both training and inference latency under identical hardware settings (A6000-48GB). For PerFit, we treat the total training cost as the sum of Stage-1 and Stage-2. As shown in the tables below, PerFit achieves **lower memory usage**, **faster inference**, and **lower training latency** relative to the strong LoRA-based method, i.e., OPPU, across all tasks.
> >
> > | Dataset | Movie Tagging | Movie Tagging | News Categorize | News Categorize | News Headline | News Headline | Product Rating | Product Rating | Scholarly Title | Scholarly Title | Tweet Paraphrase | Tweet Paraphrase |
> > | --- | --- | --- | --- | --- | --- | --- | --- | --- | --- | --- | --- | --- |
> > | Metric | Training | Inference | Training | Inference | Training | Inference | Training | Inference | Training | Inference | Training | Inference |
> > | OPPU | 31.705GB | 30.49GB | 28.765GB | 26.97GB | 27.48GB | 26.94GB | 29.355GB | 29.21GB | 31.745GB | 30.01GB | 28.545GB | 27.05GB |
> > | PerFit | 25.865GB | 25.46GB | 23.175GB | 19.91GB | 20.165GB | 18.97GB | 18.91GB | 21.51GB | 19.65GB | 21.00G | 19.34GB | 17.47GB |
> >
> > | Dataset | Movie Tagging | Movie Tagging | News Categorize | News Categorize | News Headline | News Headline | Product Rating | Product Rating | Scholarly Title | Scholarly Title | Tweet Paraphrase | Tweet Paraphrase |
> > | --- | --- | --- | --- | --- | --- | --- | --- | --- | --- | --- | --- | --- |
> > | Metric | Training | Inference | Training | Inference | Training | Inference | Training | Inference | Training | Inference | Training | Inference |
> > | OPPU | 20.59m | 8.66s | 22.86m | 37.84s | 40.19m | 30.84m | 10.51h | 17.78s | 5.87h | 8.28s | 1.50h | 17.45s |
> > | PerFit | 12.51m | 4.44s | 15.06m | 22.17s | 21.72m | 1.62m | 5.89h | 0.24s | 2.14h | 0.75s | 41.26m | 1.15s |
> >
> > ---

---

> ### Author Response · Authors · 2025-11-21
> **Rebuttal (3/3)**
>
> **Q3:** Transferability of the collective shift across users and tasks.
>
> **R:** Thank you for the question. In our setting, the collective shift learned in Stage-1 is *explicitly designed* to generalize across users: Stage-2 is always trained on **previously unseen users**, and directly benefits from the Stage-1 collective initialization without retraining or adaptation. **This demonstrates that the collective shift is not tied to the specific users seen during Stage-1, but instead captures a population-level personalization trend that transfers naturally to new users.**
>
> We also conducted a **cross-task transfer study** to examine whether the collective shift generalizes beyond the task used in Stage-1. In this experiment, Stage-1 is trained on one task (e.g., Movie Tagging), while Stage-2 is trained on a **different** task (e.g., News Categorize or Tweet Paraphrase). As shown in the table below, the transferred collective shift continues to provide stable performance across all downstream tasks. Although the performance is not as high as using the task-matched Stage-1 (since we intentionally used a unified, non-task-tuned hyperparameter setting), the collective shift remains **robust and functional** across tasks.
>
> Interestingly, this pattern aligns with our representational analysis in Appendix B.6: Tweet Paraphrase and News Headline show a large geometric separation in δ-space, which corresponds to the strongest cross-task performance drop in both directions, whereas tasks that are closer in δ-space (e.g., Movie Tagging and News Categorization) exhibit much smoother transfer.
>
> | Task | Movie Tagging | Movie Tagging | News Categorize | News Categorize | News Headline | News Headline | Tweet Paraphrasing | Tweet Paraphrasing |
> | --- | --- | --- | --- | --- | --- | --- | --- | --- |
> |  | Acc | F1 | Acc | F1 | R-1 | R-L | R-1 | R-L |
> | Movie Tagging | 0.636 | 0.520 | 0.815 | 0.591 | 0.194 | 0.174 | 0.466 | 0.420 |
> | News Categorize | 0.545 | 0.408 | 0.818 | 0.588 | 0.195 | 0.175 | 0.467 | 0.419 |
> | News Headline | 0.540 | 0.381 | 0.814 | 0.586 | 0.207 | 0.186 | 0.212 | 0.203 |
> | Tweet Paraphrase | 0.521 | 0.364 | 0.795 | 0.579 | 0.196 | 0.176 | 0.490 | 0.447 |
>
> These results show that the Stage-1 collective shift exhibits non-trivial cross-task transferability, suggesting a connection to task-level personalization characteristics rather than a single dataset. This observation is consistent with our δ-vectors analysis and points to *an interesting direction for future work on understanding when and how such collective shifts transfer across tasks by exploring benchmarks with more complex and diverse tasks.*

---

### Official Review · Reviewer_CFGx · 2025-11-03

**Soundness:** 3
**Presentation:** 3
**Contribution:** 2
**Rating:** 6
**Confidence:** 3

**Summary:**

This paper introduces PerFit, a novel parameter-efficient fine-tuning method for personalising large language models (LLMs). PerFit directly fine-tunes intervention vectors in this low-rank subspace via a two-stage process, first learning a collective shift and then user-specific personalised shifts. The method is evaluated on six tasks from the LaMP benchmark.

**Strengths:**

1. The achieved parameter reduction (81.25% to 98.44% compared to OPPU) is substantial and a clear practical advantage for scalable personalisation, where storing a unique adapter per user is a key bottleneck.
2. The paper provides a thorough experimental section, answering key research questions with main results, efficiency analysis, and detailed ablation studies.

**Weaknesses:**

1. The paper finds that intervening in earlier/middle layers (e.g., layers 5-10) works best (Fig. 6, Table 11) and that cumulative intervention can be harmful. However, the rationale for the specific choice of intervention layers is not sufficiently justified. It remains unclear if this is an optimal configuration derived from the analysis or a heuristic choice. A more principled connection between the location of the discovered low-rank subspace (Observation 1) and the optimal intervention layer would strengthen the methodology.

2. The primary baseline comparison is against LoRA-based PEFT methods. While this is relevant, the work would benefit from a direct comparison to other Representation Fine-Tuning (ReFT) methods, such as the specific method presented in [1]. The authors mention that ReFT is used as a baseline with matched parameters (L1134-1136), but its results are not shown in the main tables. A direct comparison would better situate PerFit's two-stage, low-rank approach within the emerging ReFT paradigm.

3. The concept of the "collective shift" is central to the method. However, its interpretation could be more deeply discussed. Is it primarily capturing a shift from a generic to a "personalised-in-general" mode of operation? Or could it be conflated with domain adaptation, as the collective data is task-specific (e.g., all news data for LaMP-4)? A brief discussion on disentangling these concepts would be helpful.


[1] ReFT: Representation Finetuning for Language Models

**Questions:**

1. Was the analysis in Section 3 conducted per-layer to identify which layer's representations best exhibit the low-rank property for personalisation?

2. Algorithm 1 and its context state that Stage-2 parameters are initialised from the shared parameters trained in Stage-1, rather than being initialised separately for each user. Could you elaborate on the rationale for this choice? Was ablating against random initialisation for Stage-2 parameters considered? This choice seems crucial for efficient knowledge transfer from the collective shift.

---

> ### Author Response · Authors · 2025-11-21
> **Rebuttal (1/2)**
>
> We sincerely thank the reviewer for the positive and encouraging feedback and appreciate the recognition of our parameter reduction and experimental validation. Below, **W** denotes weaknesses, **Q** denotes questions, and **R** provides our responses.
>
> &nbsp;
>
> ---
>
> **W1 & Q1:** The selection of intervention layers.
>
> **R:** Thanks for raising this insightful point. We agree that establishing a more principled connection between our representational analysis and the choice of intervention layers would be highly valuable.
>
> In our revision, we have added **a new layer-wise analytical study (Page 21, Appendix B.5, Figure 15, blue-highlighted)**, which conducts the same low-rank and collective-shift analysis from Section 3 **at every layer**.
>
> Specifically, we report the minimum feature dimension $r$ required for each layer to explain 90% of the variance below, providing a clear quantitative characterization of how low-rank structure evolves.
>
> | Layer-index | 1 | 5 | 10 | 15 | 20 | 25 |
> | --- | --- | --- | --- | --- | --- | --- |
> | **LaMP-4** | 8 | 20 | 53 | 167 | 318 | 421 |
> | **LaMP-7** | 6 | 15 | 11 | 32 | 258 | 502 |
>
> The extended analysis reveals that $\delta$-vectors in **earlier and middle layers** (approximately layers 5–15) exhibit both **a more pronounced low-rank structure** and a **clearer, more stable collective shift**—reflected in larger absolute means and lower variance—whereas deeper layers show weaker and noisier personalization patterns. This alignment between the representational properties and the empirical intervention results provides **a more principled justification** for our layer choice.
>
> &nbsp;
>
> ---
>
> **W2:** Comparison with ReFT in the main table.
>
> **R:** Thanks for the suggestion. We will move the corresponding ReFT results in Table 4 into the main tables in the updated version for better visibility. We also compared with LoFiT, another recent representation fine-tuning approach, which is already included as one of our main baselines. Across both comparisons, PerFit consistently achieves stronger or comparable performance, further demonstrating the effectiveness of our approach.
>
> |  | News Categorization |  | Movie Tagging |  | News Headline |  | Tweet Paraphrasing |  | Scholarly Title |  | Product Rating |  |
> | --- | --- | --- | --- | --- | --- | --- | --- | --- | --- | --- | --- | --- |
> |  | Acc $\uparrow$ | F1 $\uparrow$ | Acc $\uparrow$ | F1 $\uparrow$ | R-1 $\uparrow$ | R-L $\uparrow$ | R-1 $\uparrow$ | R-L $\uparrow$ | R-1 $\uparrow$ | R-L $\uparrow$ | MAE $\downarrow$ |  RMSE $\downarrow$ |
> | Ours | 0.818 | 0.586 | 0.630 | 0.518 | 0.207 | 0.186 | 0.525 | 0.472 | 0.521 | 0.451 | 0.179 | 0.443 |
> | ReFT | 0.801 | 0.594 | 0.599 | 0.473 | 0.190 | 0.171 | 0.478 | 0.433 | 0.474 | 0.411 | 0.241 | 0.509 |

---

> ### Author Response · Authors · 2025-11-21
> **Rebuttal (2/2)**
>
> **W3:** Deeper interpretation of collective shift.
>
> **R:** Thank you for the suggestion. In our framework, the *collective shift* is intended to capture a **population-level personalization trend within  task.**
>
> Since each δ-vector is defined as the representation difference with and without personalized information for the same user and query, aggregating δ-vectors across users highlights the **shared stylistic component** that is commonly introduced by personalization, i.e., **how users shift the model from its generic behavior toward a “personalised-in-general’’ mode for that task.**
>
> Our additional analyses support this view. *Appendix B.6 (Page 22, Figure 16)* shows that these collective directions form **systematic, task-dependent structures**, indicating that each task exhibits its own characteristic group-level trend. *Appendix B.4* further illustrates that the distance of a δ-vector from the collective direction reflects its alignment with this dominant pattern: δ-vectors closer to the collective shift correspond to users whose personalization styles are more similar to the group-level trend, while larger distances simply reflect greater individual deviation around this shared tendency and with similar patterns.
>
> This structure—shared patterns plus individual deviations—aligns with classic group psychology findings, where groups on similar tasks show convergent behaviors or styles [1, 2], and analytical psychology principles, where individuals express task-related shared patterns differentiated by personal traits [3]. These theories frame why stable collective shifts occur across users on the same task, with personalized layers.
>
> Together, these results show that the collective shift models the **dominant group-level personalization pattern**, whereas the personalized shift captures the remaining spectrum of individual variation. We will incorporate this clarification and the supporting analyses into the revised paper to make the interpretation more explicit.
>
> &nbsp;
>
> **References:**
>
> [1] Asch, S. E. (1956). *Studies of independence and conformity: I. A minority of one against a unanimous majority*. Psychological Monographs: General and Applied, 70(9), 1–70.
>
> [2] Sherif, M., & Sherif, C. W. (1953). *Groups in harmony and tension: An integration of studies on intergroup relations*. Harper & Brothers.
>
> [3] Jung, C. G. (1968). *The archetypes and the collective unconscious* (2nd ed., R. F. C. Hull, Trans.). Princeton University Press.
>
> &nbsp;
>
> ---
>
> **Q2:** Initialization of the Stage-2.
>
> **R:**    Thank you for raising this important question. We would like to further explain how Stage-1 and Stage-2 interact in our pseudocode. The key idea is that:
>
> - Stage-1 learns the collective shift (a fixed direction shared across users calculated through $\Delta \Theta^{(1)}$),
> - and simultaneously learns the *base location* for Stage-2’s ReFT parameters,
> - the Stage-2 personalized shift is trained *independently* and *on top of* the frozen collective shift.
>
> Initializing Stage-2 from the Stage-1–trained point simply provides a **better-oriented starting direction** inside the personalization subspace. A random initialization would still operate on top of the same collective shift, but the user-specific vector would begin from an arbitrary direction, forcing the optimizer to re-discover that subspace for every user. This does not change the functional formulation, but may make optimization noisier and less data-efficient.
>
> We ran an ablation comparing this to **random initialization**, and observed that while both variants work, **random init yields slightly lower performance**, whereas the Stage-1–aligned init provides more consistent results. This difference reflects optimization efficiency rather than a change to the modeling formulation. Our core contribution is the **two-stage representation fine-tuning paradigm**, and the initialization choice is flexible.
>
> |    | News Categorize |  | Movie Tagging |  | News Headline |  |
> | ---         | ---             | ---             | ---           | ---           | ---           | ---           |
> |      | Acc             | F1              | Acc           | F1            | R-1           | R-L           |
> | Random Init | 0.811           | 0.599           | 0.627         | 0.519         | 0.194         | 0.175         |
> | PerFit      | 0.818           | 0.586           | 0.630         | 0.518         | 0.207         | 0.186         |

---

### Meta-Review · Area_Chair_hJHc · 2025-12-26

**Summary:**

Reviewers found the paper technically solid and promising, but the remaining issues support a poster recommendation. The evaluation is still largely limited to LaMP-style personalization, so it is unclear how well the approach carries over to other personalization settings such as conversational or continuously evolving user scenarios. The meaning of the collective shift also remains somewhat ambiguous because it may reflect task or domain adaptation effects or properties of the prompt and retrieval setup rather than personalization alone. Finally, the method is not directly tested under more challenging distribution changes, for example when new user cohorts exhibit substantially different behaviors.

**Reviewer Concerns:**

The rebuttal addressed several core reviewer concerns by adding more quantitative support for the central representation claims, including layer-wise analysis that connects the discovered structure to the selected intervention layers. It also strengthened the practical case with concrete wall-clock latency and GPU memory measurements, and provided additional evidence that the representation-level interventions do not materially degrade general LLM capabilities using standard general-purpose benchmarks.

However, two major concerns remain outstanding. The empirical validation is still largely confined to the LaMP-style personalization setup, so it is unclear how well the approach generalizes to other personalization regimes such as conversational, preference-based, or continuously evolving settings. In addition, the interpretation of the collective shift remains somewhat ambiguous because it is not cleanly disentangled from task or domain adaptation effects and from potential artifacts introduced by the prompt and retrieval construction.

**Reviewer Scores:**

- CFGx (6 → 6 or 7): The rebuttal substantially reduces their main methodological uncertainty by adding layer-wise representation analysis that better justifies the intervention layers. It also clarifies the Stage-2 initialization choice with an ablation and makes the ReFT comparison more visible. Hence, I expect they would likely keep the score or raise it slightly.


 - xGBE (6 → 6 or 7): The authors more clearly distinguish PerFit from prior ReFT-style methods, supplement the shift hypothesis with additional quantitative analyses, and provide concrete training and inference cost measurements as requested. These updates should alleviate most of their stated reservations, so I expect a stable score with a possible small increase.


 - eWQo (6 → 7): The rebuttal directly answers their major requests by adding sensitivity tests for the Stage-1 user set, a task-specific explanation for Tweet Paraphrasing underperformance, detailed wall-clock and memory profiling, explicit evidence of transfer to unseen users, and general-capability evaluations. Given how closely these additions match the original concerns, I expect they would be most likely to increase their score by one point.

---

### Decision · Program_Chairs · 2026-01-26

Accept (Poster)